# Deciphering the history of ERK activity from fixed-cell immunofluorescence measurements

Abhineet Ram, Michael Pargett, Yongin Choi ⓘ, Devan Murphy,
Carolyn Teragawa, Markhus Cabel ⓘ, Nont Kosaisawe, Gerald Quon ⓘ &
John G. Albeck ⓘ ✉

The RAS/ERK pathway plays a central role in diagnosis and therapy for many cancers. ERK activity is highly dynamic within individual cells and drives cell proliferation, metabolism, and other processes through effector proteins including c-Myc, c-Fos, Fra-1, and Egr-1. These proteins are sensitive to the dynamics of ERK activity, but it is not clear to what extent the pattern of ERK activity in an individual cell determines effector protein expression, or how much information about ERK dynamics is embedded in the pattern of effector expression. Here, we evaluate these relationships using live-cell biosensor measurements of ERK activity, multiplexed with immunofluorescence staining for downstream target proteins of the pathway. Combining these datasets with linear regression, machine learning, and differential equation models, we develop an interpretive framework for immunofluorescence data, wherein Fra-1 and pRb levels imply long-term activation of ERK signaling, while Egr-1 and c-Myc indicate more recent activation. Analysis of multiple cancer cell lines reveals a distorted relationship between ERK activity and cell state in malignant cells. We show that this framework can infer various classes of ERK dynamics from effector protein stains within a heterogeneous population, providing a basis for annotating ERK dynamics within fixed cells.

The RAS signaling pathway directs multiple cellular behaviors and regulates tissue homeostasis[1]. Its terminal kinase, Extracellular Signal-Regulated Kinase (ERK), is essential for cellular decisions to enter the cell cycle, migrate, or differentiate. In cancer and other diseases, changes in the quantitative strength and timing of ERK signaling play a critical role in disease progression and treatment. For example, individual cell fates can be altered by short interruptions in ERK activity[2–4], while residual ERK activity following targeted kinase inhibitor treatment enables re-entry to the cell cycle[5,6] and determines therapeutic efficacy[7]. Therefore, accurate measurement of ERK pathway activity has substantial clinical applicability. However, current methods to assay ERK activity in situ are limited and have not been highly successful as diagnostic markers[8].

Measuring cellular ERK activity is complex because the timing, duration, localization, and amplitude of ERK activation vary widely between cells[9–12]. These differences have functional importance because they influence the expression of numerous effector proteins encoded by ERK target genes (ETGs), including the Immediate Early Genes (IEGs) *FOS* (encoding c-Fos), *FOSL1* (Fra-1), *MYC* (c-Myc), and *EGR1* (Egr-1)[13–17]. The amplitude, duration, and frequency of ERK activity interact with ETG expression by increasing transcription rate, by stabilizing protein products, and by activating negative feedbacks on expression[18]. These effects vary from gene to gene, and consequently some gene products (e.g., Fra-1) integrate ERK activity over time, whereas others (e.g., c-Fos, Egr-1) respond maximally to pulsed ERK activity at intermediate frequencies[19–21]. Correspondingly, cellular

Department of Molecular and Cellular Biology, University of California, Davis, CA, USA. ✉e-mail: jgalbeck@ucdavis.edu

phenotypes are responsive to ERK dynamics; pulsatile activation correlates with proliferation and protection from apoptosis in some systems, while sustained activity correlates with cell cycle arrest[11,22]. Advances in live-cell imaging and CRISPR-based tagging have allowed a higher-resolution view of how patterns of activation and deactivation (ERK dynamics) correlate with ETG expression[19,20]. However, at the single-cell level, the correlations in these studies appear modest, and the ETG responses are not reliably predictable. This uncertainty is not necessarily random and could arise from various features, including systemic non-linearities and unmeasured factors. Therefore, it remains important to ask whether the dynamic pattern of ERK activity in an individual cell reliably determines the expression of multiple ERK targets. Conversely, can the expression state of ERK target genes reveal information about the prior history of ERK activity dynamics in the cell? Such questions motivate the need for further quantitative analyses and modeling.

A quantitative framework connecting ERK dynamics to ETG expression states would have several notable uses. First, it would help elucidate the cell state landscape that is accessible through growth factor pathway modulation. Transcriptome-based single-cell state measurements have become widely available, but how non-transcriptional events in upstream signaling pathways drive cell state decisions and transitions remains a pressing question. Second, it would provide a comparison of how different fixed markers reflect actual ERK activity. Currently, RAS/ERK activity is assayed using antibodies for phosphorylated ERK (pERK)[7,23,24] or for proteins that are induced by ERK activity, such as c-Fos or DUSP6, which persist for longer than ERK phosphorylation[25,26]. Knowing how different markers respond to ERK activity could help in choosing the best activity marker under different circumstances and would be of particular interest in assessing the efficacy of cancer treatments targeted to the pathway. Finally, oncogene-induced ERK activation is distinct from normal physiological patterns generated by endogenous growth factors and is often sustained or stochastically fluctuating[11,27]. Knowledge of the dynamic patterns of ERK signaling found in a tissue can therefore be informative about the source of ERK hyperactivation.

In addition to ERK activity biosensors, previous work has demonstrated that synthetic ETGs can detect ERK dynamics[28]. While such genetically encoded tools are impractical clinically, it is possible that similar information could be derived from endogenous ETGs, which range in their sensitivity and timing in response to dynamic ERK activity[20,29]. In principle, the history of cellular ERK activity could be inferred using fixed-cell measurements alone and used to evaluate the dynamic nature of ERK activity within biopsies of tumor tissue.

Here, we develop such an interpretive framework, using a live-cell biosensor of ERK activity in combination with cyclic immunofluorescence for ETG-encoded proteins and other proteins regulated by ERK, including Egr-1, Fra-1, c-Jun, c-Myc, and c-Fos, and phosphorylated proteins such as pERK, pc-Fos, and pRb (a downstream marker of ERK-dependent cell cycle entry; for convenience we collectively refer to all of these measurements as ETGs). Because of the well-established interdependence of protein stability and ERK-mediated phosphorylation that modulate ETG expression[15], we hypothesized that this panel would provide the best assessment of ERK activity available through immunofluorescence. Using statistical models and machine learning to predict ERK activity features based on the expression of each marker, we characterized how various ETGs report ERK history with varying memory span. The characteristics of Fra-1 and pRb as long-term integrators and Egr-1, c-Fos, or c-Myc as short-term responders provided significant predictive capacity in our models. We further expanded on these models by experimentally examining multiple cancer cell lines and by computationally simulating ERK-driven gene expression, demonstrating that fixed-cell analysis of ETGs can be broadly useful in revealing the dynamic history of ERK activity.

## Results

### A dataset linking live-cell ERK activity to ERK target immunofluorescence

We generated a dataset that enables correlation of ERK activation to downstream target expression and modification by first collecting live ERK activity measurements under differential activation states of the RAS/ERK pathway. In MCF10A mammary epithelial cells, we used EKAR3.1, a calibrated FRET-based biosensor that measures the balance between phosphorylation by ERK activity and dephosphorylation by competing phosphatases (Fig. 1a, Supplementary Fig. S1a–d). ERK activity was stimulated in a dose-dependent manner with varying Epidermal Growth Factor (EGF) concentrations (Fig. 1b, S1e). To increase the temporal diversity of activity patterns, we added MEK inhibitor (MEKi) at varying times after EGF stimulation and included treatments where EGF was added at different timepoints of the experiment (Fig. 1c, Supplementary Fig. S1e, Supplementary Table 1). These treatments generated a wide range of ERK signaling patterns, including sustained and pulsatile activity with varying duration and magnitude (Fig. 1d). Consistent with previous studies[19,30], we found that ERK activation is heterogenous from cell to cell within each stimulation condition.

Immediately following live-cell data collection, we fixed the cells and conducted cyclic immunofluorescence staining to measure levels of eight targets downstream of ERK (Fig. 1a, e, Supplementary Movie 1). This protocol (4i) was adapted from Gut et al.[31] and validated for our 96-well plate experiments (Supplementary Fig. S2a–e). After quantifying nuclear antibody staining intensities, we found that most targets were dose-responsive to EGF and suppressed by MEKi treatment, except c-Jun (Fig. 1e, f, Supplementary Fig. S1e). c-Jun increased moderately with both MEK inhibition and EGF concentration, suggesting that its expression is not directly regulated by ERK activity in MCF10A.

We then analyzed the correlation between ERK activity and the expression of each target. To link live-cell ERK activity measurements with the respective 4i data for each cell, we aligned the corresponding image datasets and generated a heatmap ordered by the mean ERK activity measurement in each cell (Fig. 2a). While both ERK activity and 4i targets were variable across the data set, most of the 4i targets exhibited some correlation with mean ERK activity. We calculated the Pearson correlation between ERK pulse features, such as Frequency, for each cell and each ETG measurement (Fig. 2b, c). The strongest correlations were between the Sum of duration and Fra-1 or pRb. Interestingly, Egr-1 was uniquely correlated with Average derivative of ERK activity, supporting the previous notion that Egr-1 selectively decodes pulsatile ERK activation[21]. Notably, c-Jun had a weak correlation with any ERK feature and therefore serves as a useful negative control in our analyses, providing baseline correlation values for proteins unregulated by ERK.

We performed a more granular time-sensitive analysis by calculating the Pearson correlation between each ETG and the EKAR3.1 FRET measurement at each live-cell timepoint (Fig. 2d). Correlations to Fra-1 and pRb were distributed across the time series (r = -0.5, 0.4, respectively) after the initial stimulus, apart from a period where ERK activity is weakest, about 2 to 4 hours after EGF addition. In contrast, c-Myc, c-Fos, and pc-Fos mildly correlated to ERK activity from 2–5 hours prior to fixation (r = -0.3) and at their highest levels (r = -0.55) during the last hour before fixation. pERK staining was most correlated to ERK activity immediately prior to fixation (r = -0.6), consistent with its rapid time-varying nature. Finally, Egr-1 correlated only to ERK activity 30 minutes to 1 hour prior to fixation (r: -0.5).

To visualize spatial correlations of ERK-ETG signaling within the dataset, we plotted a spatial heatmap of signaling and gene expression, where cells within a single image are clustered in a heatmap visualization by proximity to one another (Fig. 2e). This analysis shows ERK activation within clusters of neighboring cells (5–30 cells) throughout the experiment. Consistent with its correlation to recent ERK activity (Fig. 2d), Egr-1 expression is higher in these clusters of cells subject to

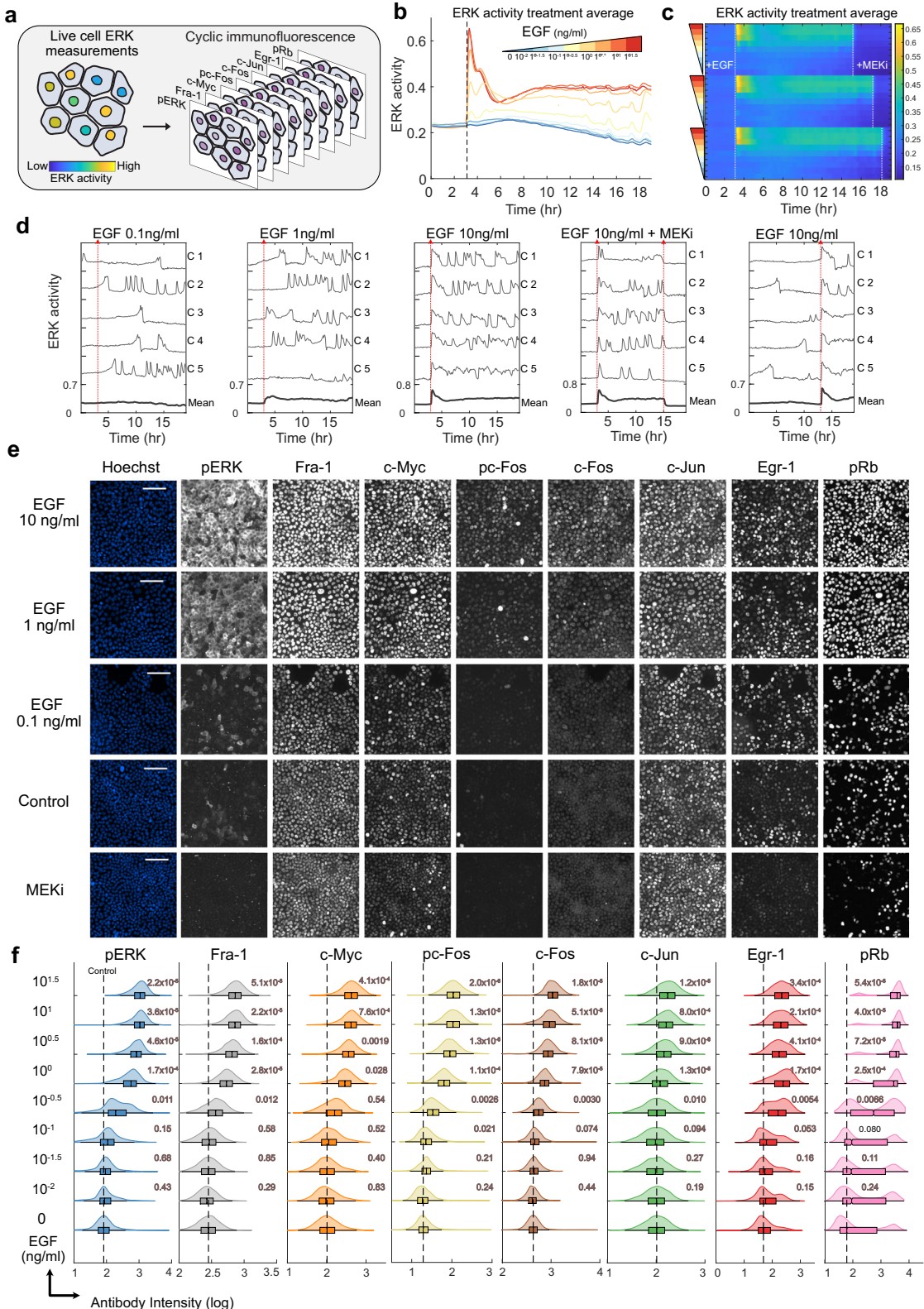

locally synchronized ERK activity bursts near the end of the time-lapse imaging phase (Fig. 2f).

## Neural network-based models reveal non-linear time dependence of ETGs on ERK dynamics

To look beyond linear correlations and identify potential non-linear or complex relationships between live ERK activity and ETGs, we trained a convolutional neural network (CNN) to use the ERK activity time series to predict expression levels of each ERK target in individual cells (Fig. 3a, top). As a comparison to the CNN, we fitted linear regression models using the values at each timepoint of the ERK time series as individual variables to predict final ETG levels (denoted TS linear). We also compared the performance of these time series-based models with that of feature-based models, using nine ERK activity features (as

**Fig. 1 | ERK activity and target genes are dose-responsive to Epidermal Growth Factors. a** Schematic of the experimental method. Live cells were imaged in 96-well plates for 19 hours and immediately fixed following the end of the time-lapse measurements. Plates were subsequently stained for antibody-based measurements. **b** Condition average response measurements for live-cell ERK biosensor (EKAR signal, shown in arbitrary units) with increasing concentrations of EGF. Data are presented as the mean of each condition ($n_{replicates}$ = 3 for all conditions). **c** Condition average response measurements depicted as a heatmap. Each row represents the average EKAR FRET measurement for a condition, indicated by the color scale (yellow, high ERK; blue, low ERK). EGF concentration is indicated by colored triangles as in Fig. 1b. MEKi = MEK inhibitor PD0325901 (100 nM) (see Supplemental Table 1 for $n_{replicates}$ for each condition). **d** Plots of single-cell EKAR

signals (in arbitrary units) for five representative cells in each indicated condition. Bold lines indicate the mean of all cells in one well of the condition. Red lines indicate the time points where treatments were added. **e** MCF10A cells immunostained with cyclic immunofluorescence. Each row depicts the same group of cells. Scale bar = 100 μm. **f** Nuclear quantification of cyclic immunofluorescence measurements from listed EGF condition. Plots indicate the median (bar), 25th/75th percentiles (box), and range of the data (whiskers). The dashed line indicates the median of vehicle control (Imaging Media). Variance-corrected t-tests (two-sided) were conducted by comparing each EGF-treated condition to control. $n_{replicates}$ = 3. *P* values, relative to the 0 EGF condition, are shown on each distribution. Throughout the study, $n_{replicates}$ is used to denote independent experimental replicates performed on different days.

shown in Fig. 2b; denoted Featurized linear). We found that the CNN achieved the highest performance in predicting all ERK targets, except for pERK (Fig. 3b). To account for overfitting, we calculated the mean squared error (MSE) on unseen data (test set); the CNN exhibited the least error for all targets, again except for pERK (Fig. 3b bottom). Though the CNN performed best for most targets, significant variance remained uncaptured (Fig. 3c). The improvements in variance explained were significant only for Egr-1 and pRb, suggesting nonlinear responses to ERK activation. For many 4i targets, the featurized linear models underperformed both methods in $R^2$ and test set error, showing that a priori featurization can miss key signal aspects.

We then used CNN model parameter weights (feature importance) to identify specific timepoints influencing target expression. Initially, we found that feature importance was overshadowed by the initial stimulus response, likely reflecting cellular biases rather than direct biochemical regulation of ETGs (Supplementary Fig. S3). Therefore, we limited the model to timepoints greater than 5 hours after treatment, with minimal decrease in CNN performance. A CNN trained on these time points was broadly consistent with the linear Pearson correlation analysis (Fig. 2d), but with less disperse feature importance spread over time, especially in the case of pRb (Fig. 3d). The CNN for Fra-1 expression is influenced by a broad time span, peaking 12 hours prior, with minimal effect from the last two hours. c-Fos and pc-Fos are influenced over six hours but focus on the last two to four hours. The Egr-1 model is strongly affected by ERK activation in the last two hours, while pERK, c-Myc, and pRb are influenced mainly in the final hour. This approach confirms that each ETG has a distinct sensitivity to ERK timing, with a non-linear relationship in some cases.

## Regression modeling provides backward inference of ERK dynamics

While the preceding models consider whether ERK dynamics can predict the expression strength of ETGs, a model of the reverse relationship – i.e., inference of ERK dynamics based on the pattern of ETG expression – would be of significant value because live-cell measurements are often unavailable. To explore such models, we performed cross-validated linear regression using the 4i ETG measurements as predictors and FRET-measured dynamic ERK features as output variables. We restricted our analysis to our existing feature set (Fig. 2b), which provided a convenient and intuitive means to develop such models, but our approach would remain applicable to other featurizations of ERK activity dynamics. Our analysis considered both single ETGs as predictors (Fig. 4a, b) and multiple linear regression (MLR) models using all 4i measurements from each cell (Fig. 4a, c). The top single ETG predictor based on variance explained ($R^2$) was Fra-1 for Sum of ERK duration ($R^2$ = 0.42). MLR models improved the predictions for all features, with the Sum of duration still the best-predicted feature ($R^2$ = 0.53). In comparison, amplitude characteristics of ERK activity, like Max or Average peak height, and features related to pulsatile ERK behavior, such as Average duration and Frequency, were poorly predicted by individual parameters ($R^2$ < 0.15) or by MLR ($R^2$ < 0.2).

For all ERK features, we found that near-maximal $R^2$ values could be achieved with 2 to 3 predictor ETGs (Fig. 4d, Supplementary Fig. S4). Fra-1 and pRb were the primary contributors to aggregate-based features (Mean, Sum of duration, Sum of peak height), and Egr-1 and pRb the main contributors to dynamics-related features (Average derivative, Average inter-pulse interval, and Frequency; Fig. 4d). Effectively, Fra-1, Egr-1, and pRb alone capture >90% of the predictivity achieved by the full ETG panel in this dataset. Pairwise correlations between ETGs indicated shared mutual information (Supplementary Fig. S4c), explaining why relatively few are needed for optimal models.

While $R^2$ values for single-cell models were modest, strong predictions of all features were achieved when using the average ETG values for each condition to predict average features (Fig. 4e, f); $R^2$ values ranged from 0.72 to 0.98 across MLR models of all features. Fra-1 and pRb were consistently strong individual predictors for all features ($R^2$ > 0.7) except Frequency, where Egr-1 was the top predictor. While Frequency remained the most difficult feature to predict, its MLR $R^2$ rose from 0.08 at the single-cell level to 0.39 in the average model. Overall, the population average models reconcile the modest predictive power of the single-cell models and confirm the classical view that, on average, ERK strongly determines ETG expression levels.

Notably, pERK performed poorly as a predictor of ERK activity, in both forward (Fig. 2) and reverse (Fig. 4) modeling approaches. This weak relationship arises partly from conditions where MEK inhibition following EGF quickly diminishes the pERK signal (see Fig. 1). Removing these treatments improved regression models for pERK and slightly for other 4i measurements. (Supplementary Fig. S4d). These results indicate that pharmacological inhibition renders pERK an unreliable predictor of ERK activity histories and that relying solely on pERK staining can lead to misinterpretations of pathway activation. In contrast, Fra-1 and pRb measurements remain robust to MEKi treatments and better predict long-term ERK activation.

## Cancer cell lines differ in the relationship between ERK dynamics and ETG state

To establish the generality of the modeled relationships, we next expanded our investigation to additional cell types, multiple mitogenic stimuli (EGF or bovine serum), and additional candidate ETG stains. We generated stable MCF7 (breast adenocarcinoma), HCC827, and A549 (both lung adenocarcinoma) cell lines carrying EKAR3.1. These lines were subjected to a similar live- and fixed-cell analysis. Additional immunofluorescence stains, including phospho-EGFR, phosho-4EBP1, phospho-p70S6K, NF-1, DUSP1/4, EZH2, FoxO1, RSK1, GSK3β, and E-Cadherin were added, to expand the panel to 19 signaling-related proteins (Supplementary Figs. S5–S7, Supplementary movies 2–4). We first noted distinct overall ERK activity patterns for each of the cell lines. HCC827 cells featured high average ERK activity regardless of EGF stimulation conditions, consistent with their EGFR mutant (E746-A750 deletion) status (Fig. 5a). This activity was highly variable and disorganized at the single-cell level (Fig. 5b). In KRAS-mutant (G12S) A549 cells, ERK activity had a higher baseline relative to full inhibition and was more responsive to EGF, but individual cells lacked pulsatile

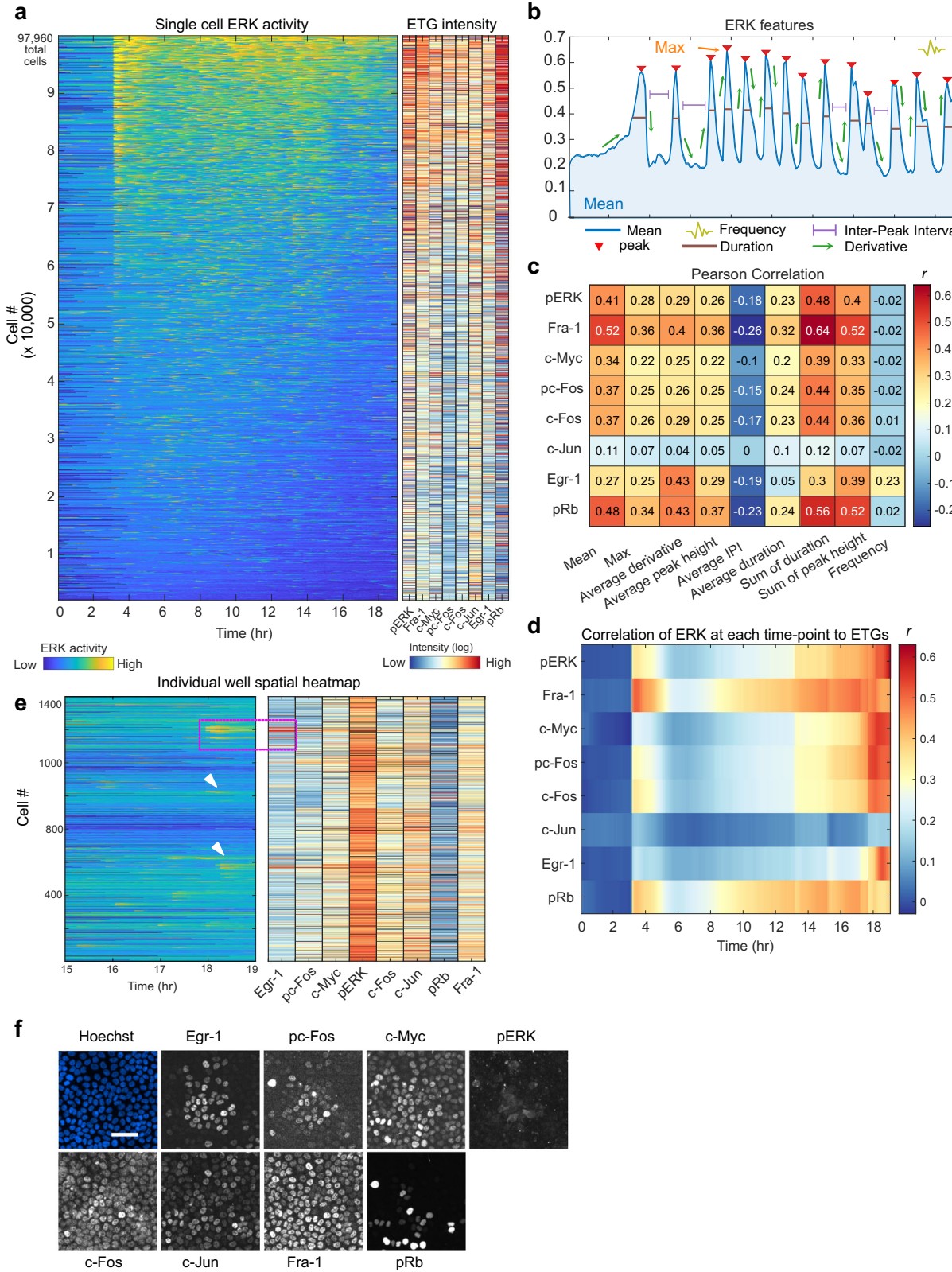

ERK activity, unlike the other cell lines (Fig. 5a, b). MCF7 cells (estrogen receptor positive, PIK3CA mutant) showed a strong but brief response to a high concentration of EGF and a more sustained response to lower EGF in the population mean (Fig. 5a). In individual MCF7 cells, ERK activity occurred in discrete pulses, similar to MCF10A cells. Overall, this panel of cells provides a diverse set of ERK activity profiles that widens the scope of gene regulatory behaviors to be observed.

In general, ETG correlations with ERK activity time points (*r*) were lower in cancer cells than in MCF10A cells (Fig. 2d), with MCF7, in particular, showing no correlations above 0.15 (Fig. 5c). However, a number of features remained similar to the MCF10A dataset; Fra-1 was among the ETGs most highly correlated with ERK activity over a wide range of time points for HCC827 and A549 cells (Fig. 5c). Also as in MCF10A, for all 3 cell lines, Egr-1, and pERK were most correlated with

**Fig. 2 | ERK target gene expression moderately correlates with features of ERK dynamics. a** Single-cell heatmap for EKAR FRET measurements and corresponding ETG intensity, each row represents one cell ($n_{cells}$ = 97,960, $n_{replicates}$ = 3). ETG expression colored by intensity of immunofluorescence (IF) measurements ($\log_{10}$). **b** Schematic of ERK dynamic features used for analysis. Frequency was calculated by estimating the mean normalized frequency of the power spectrum of the EKAR FRET measurement time series for each cell. **c** Pearson correlation (r) between each ERK feature and each cyclic IF measurement, where single-cell values were used. **d** Pearson correlation (r) between single-cell ETG measurements and the EKAR FRET measurement at each timepoint from the live-cell experiment. **e** Spatial heatmap of EKAR (left) and ETG (right) measurements from a single well (control

condition). Heatmap is organized by proximity of cells to each other so that neighboring cells in the well are plotted closer to each other in the heatmap. ETG colormap indicates the relative log intensity of data within each column; outliers in pERK column skew colormap towards red. (black = NA). Magenta box indicates cells pictured in **f**. White arrows indicate additional cells that recently activated ERK, which resulted in higher Egr-1 expression (right). **f** Images corresponding to the cells plotted within the magenta box in e, representing an example of an association that was observed consistently across all 3 experimental replicates. All panels shown are registered images of the same cells, with the scale bar indicating 50 μm shown in the Hoechst-stained image.

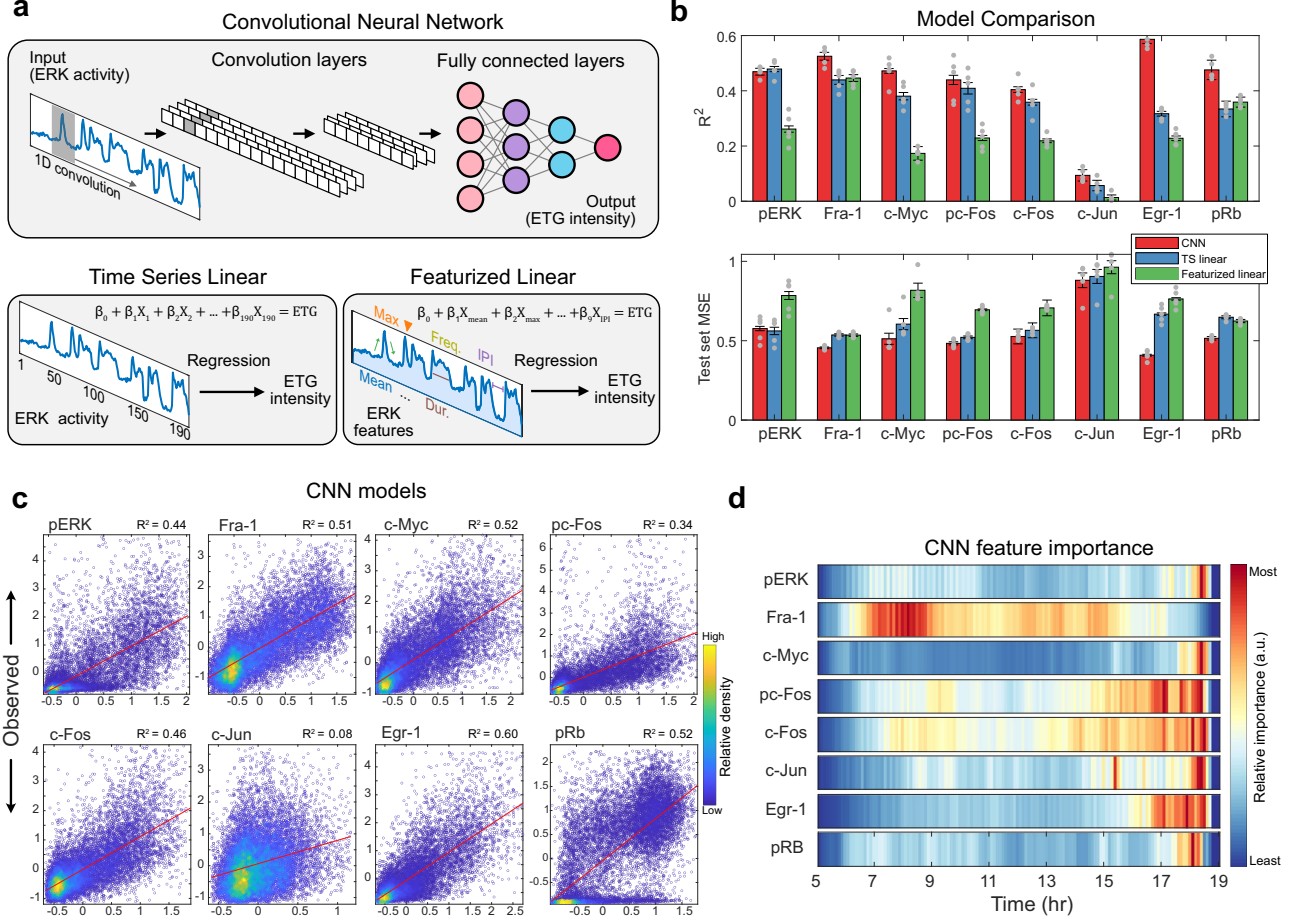

**Fig. 3 | Convolutional neural network identifies non-linear signal transmission. a** For each ETG, three types of prediction models were separately trained. Top: Simplified schematic of convolutional neural network architecture containing two convolutional layers and three fully connected layers. Bottom left: Multiple variable regression where ERK activity at each time point is considered as a predictor variable (TS linear). Bottom right: Multiple variable linear regression where nine features of ERK activity are considered as predictor variables (Featurized linear). **b** Top: Bar plot indicating $R^2$ for three models used to predict ETG levels. Bottom:

Bar plot indicating mean square error for three models used to predict ETG levels. Error bars represent standard error calculated using values from each fold of the 5-fold cross-validation partitions. **c** Scatter plot of the predicted and observed values of the CNN trained on all 190 timepoints (19 hr). The data represent standardized (z-scored) values. **d** Feature attribution heatmap showing the importance of each timepoint in the CNN model trained on 150 timepoints (15 h). Colormap represents relative values within each row. **c** and **d** represent the validation set of the first 5-fold partition.

ERK activity in the 1.5 hours prior to fixation. Some cell line-specific differences were noted; pc-Fos showed elevated correlations across the time points only in HCC827, while pp-70S6k showed a strong late time point correlation in A549. Unlike in MCF10A, some ETGs such as c-Myc and c-Fos showed very weak time series correlations with ERK activity in all three cancer cell lines, while c-Jun showed a higher degree of correlation only in A549. Additional features emerged as moderate correlates of long-term ERK activation when per-condition averages were considered, such as Fra-1, RSK1, and DUSP6 in MCF7 cells

(Supplementary Fig. S8). Interestingly, pRb showed a low correlation with ERK activity across all conditions for all three cancer cell lines, despite being one of the strongest correlates with ERK activity in MCF10A cells.

Consistent with the lower correlation values observed for the cancer cell lines, regression models of ERK activity features based on ETG measurements were somewhat less predictive of cell-to-cell variance than the corresponding models for MCF10A (Fig. 5d, Supplementary Fig. S8a). While averaging values for each condition improved

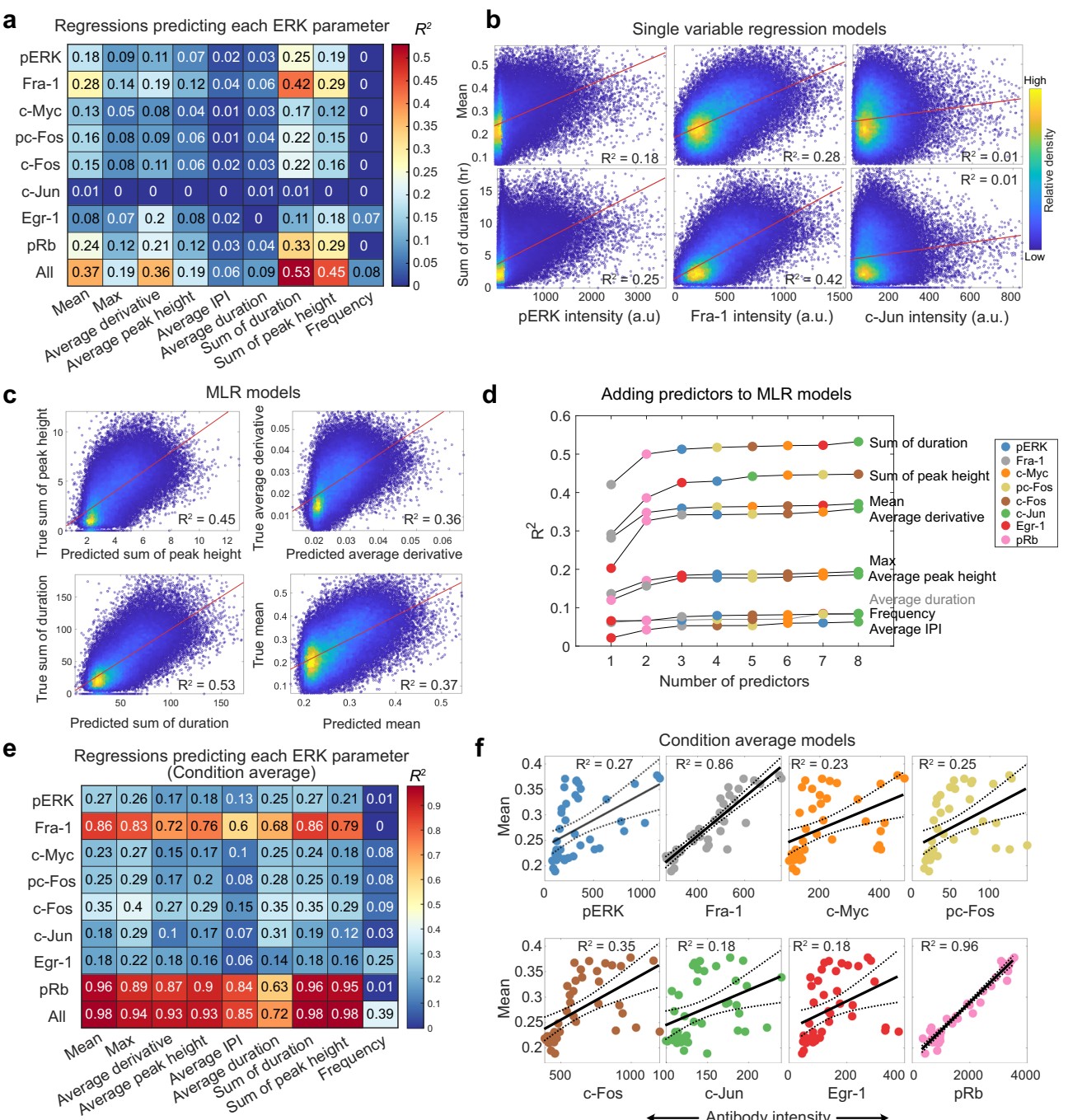

**Fig. 4 | ERK target gene expression predicts history of ERK activation. a** Single-cell regression showing the coefficient of determination ($R^2$) of linear regression models that use ETGs to predict each ERK feature. 10-fold cross-validation was conducted to retrieve the best test-set model. This model was then fit on the full dataset. "All" indicates multiple regression models using all ETGs as predictors. **b** Scatter plots of single-cell regression models showing line of best fit. Color indicates relative density of the data. **c** Scatter plot showing each cell's predicted (x-axis) vs true (y-axis) value in the multiple linear regression (MLR) models. **d** Results of adding predictors to MLR models. Color of each point indicates which predictor was added at each step. **e** Average values were calculated for all cells with the same condition. These values were then used to fit regression models that predict each ERK feature using ETGs. **f** Scatter plots showing line of best fit and confidence intervals for condition average regression models. Each dot indicates the average of one condition.

predictions (Supplementary Fig. S8c), correlations remained below those observed for MCF10A cells. Altogether, while some dynamic features of ETG induction by ERK are conserved across the cell lines, there is a strong general trend toward lower correlations and significant variation among cell lines as to which features are most highly correlated. We interpret these differences as an indication that in the tumor cell lines examined, ERK is active but its linkage to downstream regulation of gene expression and other processes is dysregulated.

Interestingly, we also noted that while the cell cycle marker pRb is well correlated with the ERK-related markers c-Myc and Fra-1 in MCF10A (Supplementary Fig. S4c), pRb is correlated strongly with c-Myc but poorly with Fra-1 across all three cancer cell lines (Supplementary Fig. S8d). Given that Fra-1 remains the best-correlated marker of ERK activity in these lines, this result suggests that cell cycle regulation has become unlinked from ERK activity in these cells. In accord with this the percentage of pRb-positive cells, a reliable indicator of cell cycle

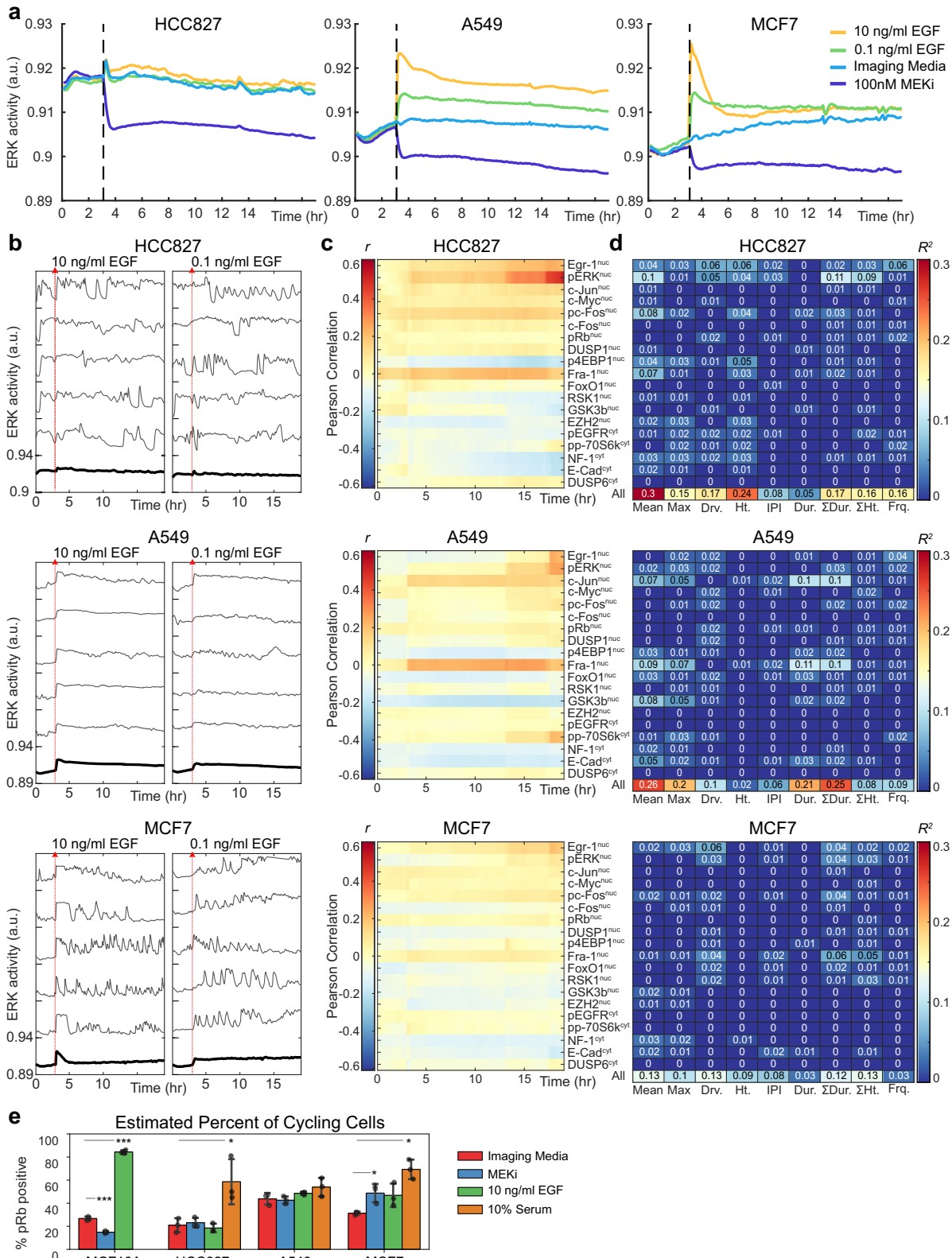

**Fig. 5 | Cancer cell types display deficiencies in processing ERK dynamics.**
**a** Condition average responses for live-cell ERK biosensor (EKAR) under four conditions. Data are presented as the mean of each condition (n_well replicates = 3). **b** Single-cell response plots to indicated condition. The bold line indicates the average of all cells in one well of the condition. **c** Pearson correlation (r) between single-cell protein measurements and the EKAR FRET measurement at each time-point from the live-cell experiment. **d** Single-cell regression showing the coefficient of determination ($R^2$) of linear regression models that use protein levels to predict

each ERK feature. 10-fold cross-validation was conducted to retrieve the best test-set model. This model was then fit on the full dataset. "All" indicates multiple regression models using all proteins as predictors. **e** Percentage of cells classified as pRb-positive or pRb-negative; intensity thresholds identified individually for each cell line. Independent t-tests (two-sided) were conducted by comparing each condition to the Imaging Medium control. n_replicates = 3. p-vals (left to right): 0.000527, 0.000002, 0.033124, 0.020208, 0.001571. Error bars: standard deviation; a.u., arbitrary units.

progression, was insensitive to MEK inhibition in all three cancer cell lines (Fig. 5e). Further, using UMAPs to visualize ETG expression across cell lines, we observed that cycling cells (pRb-positive) consistently clustered together (Supplementary Figs. S2f, S5e, S6e, S7e). Among the ETGs, c-Myc was uniquely associated with cycling cells, while other ETGs were variably expressed in both cycling and non-cycling populations. The retained correlation of pRb with c-Myc may reflect the role of c-Myc in translation regulation, as protein synthesis rate is a key factor regulating cell cycle entry independently of ERK activity[2].

## Classification models uncover prototypical patterns of ERK signaling with distinct gene expression profiles

We next explored how the combined live-cell/immunofluorescence datasets could be used to augment immunofluorescence-only datasets, by developing a method to provide concise visual annotations of inferred ERK activity history for immunofluorescence data. We first used k-means clustering to group cellular ERK activity time series into similar response classes, or prototypes (Fig. 6a). To facilitate intuitive annotations, we chose five clusters, which represented moderate activity with recent inactivation (cluster 1), consistently low activity (cluster 2), recent activation (cluster 3), mid-term activation (cluster 4), and high long-term activation (cluster 5). Analysis of the ETG expression levels in each cluster was consistent with our previous statistical models (Fig. 6b). Most ETG stains were highly expressed in clusters 3 and 5, and to a lesser extent in cluster 1, with varying enrichment across stains, offering potential for class distinction. Clustering in cancer cell lines revealed differential expression in pERK, pp-70SK6, and other ETGs, which could provide a basis for discriminating similar clusters in these lines (Supplementary Fig. S9).

We next trained ensemble-based classifiers to predict these prototypes of ERK signaling history using ETG staining levels (Fig. 6c, d). The overall prediction accuracy of our model was ~60% (compared to 20% for random selection), while individual class predictions varied in accuracy. Long-term high activity class predictions were the most accurate (83%), and moderate activation classifications were the least accurate (33%). The classifier's residual confusion reflects poor separation of some classes due to wide intra-class variation in ERK activity among individual cells (Fig. 6c). In this classification, pRb and pc-Fos were found to be the most important predictors, followed by Fra-1 and cMyc; Egr-1 was of relatively low importance (Fig. 6d). This result indicates that while pc-Fos may not explain a high amount of variance in the ERK history, it carries particularly useful information for distinguishing among the five classes identified here.

Because our other models performed more effectively on averages of multiple cells (Fig. 4e), we asked whether using ETG measurements from groups of adjacent cells could improve the prediction of ERK histories. We grouped cells within each image into hexagonal regions (radius = 50 μm; each containing from ~5 to ~30 cells) and calculated the average ERK activity time course and ETG expression values for each region. We then repeated the generation of clusters and training of classification models. To carry this approach to its practical limit, we also performed the same analysis on whole images (500-1000 cells). We then compared models generated with each approach, across all four cell lines (Fig. 6e). For all cell lines, the regional model improved model accuracy, in some cases dramatically (e.g. from 59% to 79% for MCF10A or 31% to 56% for MCF7 cells). Whole-image models, however, did not consistently outperform regional models, likely due to training on only a small number of samples, when each image only provides one data point. Finally, we overlaid markers of the spatiotemporal histories inferred by the single-cell or regional models on images from the immunofluorescence dataset (Fig. 6f). These images demonstrate both variegated spatial distribution of ERK of history and inhibitor-driven shifts in predominant ERK profiles.

## Dynamical systems modeling of ERK-driven gene expression

To investigate the theoretical limits of predicting ERK dynamics from ETG levels, we extended an ordinary differential equation (ODE) model representing the regulation of ETGs[19,29] (Fig. 7a). For a given ERK activity time series, the model simulates the mRNA and protein levels of a hypothetical ERK-responsive gene (sim-ETG). We constructed 1000 hypothetical sim-ETGs by randomly assigning each one with different values for 6 critical parameters: mRNA degradation rate, protein degradation rate, phosphorylated protein degradation rate, protein dephosphorylation rate, negative feedback half-max concentration, and fractional expression at baseline (Fig. 7c, Supplementary Table 2). These 1000 configurations survey the parameter space, allowing us to identify sim-ETGs that capture different aspects of ERK signaling. Using 10,000 randomly selected live-cell ERK activity measurements from our experimental data, we simulated responses of all 1000 sim-ETGs for each cell (Fig. 7b, Supplementary Fig. S10a). The end point sim-ETG protein values (representing a fixed-cell 4i measurement of the hypothetical protein) were examined with single variable regression modeling to characterize each sim-ETG's capacity to predict ERK dynamics features. We found that 49% of sim-ETGs could predict average ERK activity with an $R^2$ above 0.5, and more than 1% were excellent predictors ($R^2 > 0.8$) (Fig. 7d, Supplementary Fig. S10b). However, only 12% of sim-ETGs could predict maximum activation or average pulse height with an $R^2$ above 0.5, with a maximum $R^2$ around 0.6 (Supplementary Fig. S10c). Models for predicting dynamic ERK features like Frequency or Average derivative were overall worse than integrative features like the mean or sum of duration, reflecting that sim-ETGs under this model are variations on an integrator of ERK activity (Supplementary Fig. S10c).

To visualize simulated gene expression responses, we plotted a single cell's ERK signal along with the response of the top five predictors of the mean (Fig. 6f), which included both genes activated by and inhibited by ERK. While our experimental ETG measurements were selected based on known positive responders to ERK, 20% of sim-ETGs were negatively regulated by ERK (Supplementary Fig. S10b); therefore, experimental prediction of ERK activity would likely be improved by including genes that are inhibited by ERK[32]. We then analyzed which parameters most influence how well an individual sim-ETG predicts mean ERK activity by examining the weights from an MLR model of sim-ETGs (Supplementary Fig. S10d,e). We found that low mRNA and phosphorylated protein degradation rates were generally associated with accurate recording of average ERK activity, which is consistent with the measured parameter values and behavior of Fra-1[19].

We next used sim-ETGs to examine the distinction between genes that reflect long-term history (e.g. Fra-1) and those responsive to recent ERK activity (e.g. Egr-1 and c-Myc). We calculated the correlation between the ERK activity at each timepoint and end-point protein expression (analogous to the experimental data in Fig. 2e). As expected, genes predicting mean ERK activity are correlated with ERK activity over a broad time span, behaving like Fra-1. Genes are less effective at predicting mean correlate with recent activation, similar to Egr-1 or c-Myc (Fig. 7e). Notably, No sim-ETG in this model specifically predicted intermediate activation timescales (5-10 hours before fixation), indicating that such behaviors are rare in the system topology we studied.

Finally, we investigated how increasing the number of ETG measurements affects the ability of MLR models to predict ERK activity features (Fig. 7g). Overall, the predictability of ERK activity features showed similarities to our experimental findings. Some features (Mean, Sum of duration) were well predicted with only a few sim-ETGs, while others (Average inter-peak interval, Average duration) were poorly predicted regardless of the number of sim-ETGs. However, in some cases (Average derivative, Frequency), high predictability was feasible, but only with a larger set of 5 – 20 sim-ETGs (Fig. 7g inset). These results were not obtained through overfitting, as the test set

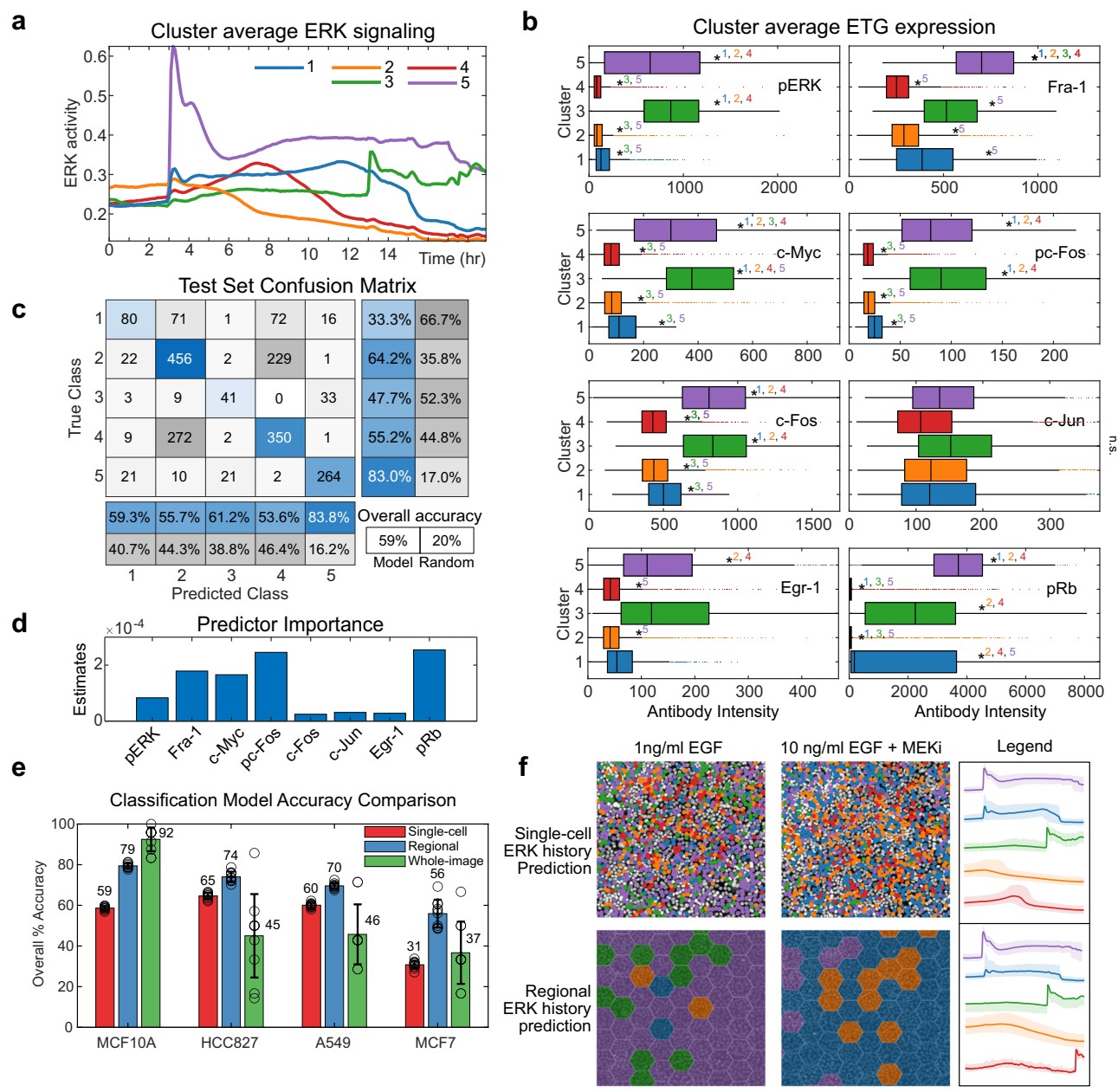

**Fig. 6 | Inferring spatiotemporal ERK patterns using classification models.**
**a** Average ERK activity in each cluster identified by k-means clustering of EKAR time series data in single cells. **b** Box plot showing median, 25th/75th quartiles, and range of ETG intensity in each cluster. One-way ANOVA test was conducted to compare the means of each group to each other. $n_{replicates} = 3$. *pval < 0.05, compared to indicated group(s). p-values indicated in Sup. Table 4. **c** AdaBoostM2 algorithm was trained to predict the cluster ID of each cell using its ETG measurements as predictors. The model with the best test-set performance, using 10-fold cross-validation, is shown. **d** Predictor importance estimates of each ETG in the model shown in **c**. **e** Comparison of model performance as a function of region size. Red: Single-cell models (as shown in **a**–**d**) for each cell line. Blue: Models for hexagonal regions (radius = 50 μm) of cells, where clustering and AdaBoostM2 models were

performed on average ERK signaling and ETG expression values for each region. Green: Models generated using the entire image (702 μm by 785 μm), using a similar training procedure, except a standard decision tree was used due to the low sample size ($n_{samples}$: MCF10A$_{242}$, HCC827$_{71}$, A549$_{72}$, MCF7$_{68}$). Error bars: standard dev. of test-set accuracy across 10-fold cross-validation. Center of error bar: mean. MCF10A models were trained with 8 predictors; cancer cell models were trained with 19 predictors. **f** MCF10A images (Hoechst) overlaid with inferred signaling histories using single-cell (top) or regional (bottom) models. Left: Cells treated with EGF for 16 hours. Right: Cells treated with EGF for 12 hours, then treated with 100 nM MEKi for 4 hours. Some cells remain unlabeled due to incomplete predictor data. Legend: Bold lines indicate mean ERK activity in observed clusters (as in Fig. 6a); shaded regions indicate 25th and 75th percentiles.

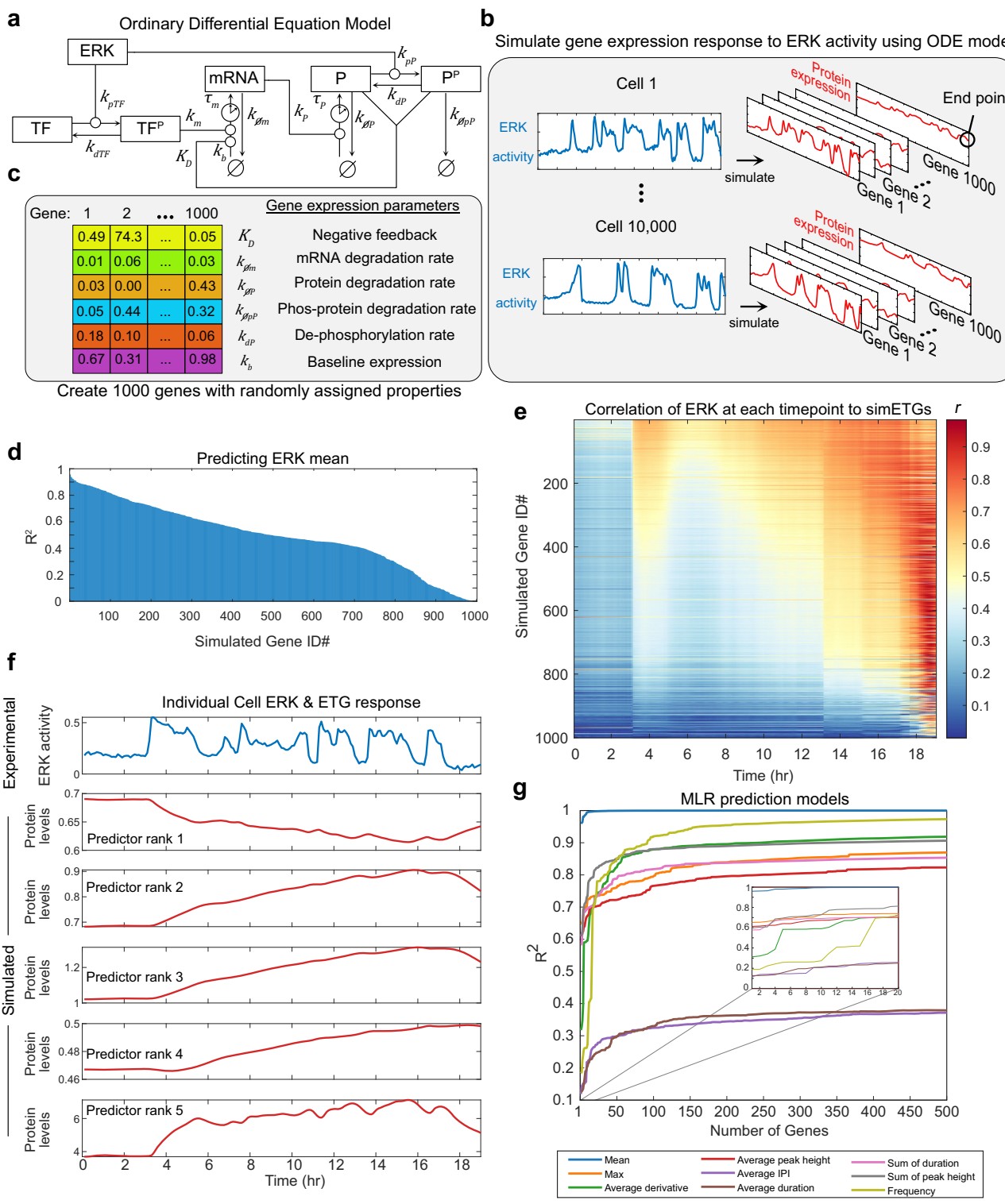

**Fig. 7 | Mathematical model identifies limits of ERK activity prediction method. a** Ordinary differential equation model representing ERK-dependent modification of a transcription factor (TF), expression of mRNA, and expression of a protein (P) product. Superscript P denotes phosphorylation of a molecule. Lowercase k's indicate rate parameters, uppercase K indicates a dissociation constant for feedback effects. Clock icons indicate a time delay ($\tau$). **b** 10,000 cells were randomly picked from our main dataset. We simulated the gene response of 1000 genes for each cell using our experimentally collected EKAR measurements. **c** 1000 simulated genes (sim-ETGs) were randomly assigned the listed rate parameters while other rate parameters in the model remained constant.

**d** Coefficient of determination ($R^2$) of single variable linear regression models using each sim-ETG to predict the average ERK activity in each cell. **e** Pearson correlation (r) between each sim-ETG (rows) measurement and the EKAR values at each timepoint from the live-cell experiment. **f** Top: one ERK activity trace from one cell in the dataset. Rest: gene expression response of the top 5 predictors of the mean ERK activity. **g** Multiple regression models fit to predict each ERK feature. For each ERK feature prediction model, a single sim-ETG was added as a predictor at each step. To determine the order of sim-ETGs to add, we performed single regression and ranked sim-ETG by their ability to individually predict each feature.

error decreased with more sim-ETGs (Fig. S10f). Altogether, these simulations suggest that predictions of some of the features in our experimental models (e.g. Mean, Max, Average peak height; see Fig. 4a), underperform their theoretical counterparts and could be substantially improved with just one or two additional ETG antibodies well suited for those parameters. Other features such as Average peak duration will likely not improve greatly even with additional stains. Average derivative and Frequency predictions could potentially be improved, albeit with the considerable investment of identifying at least several additional antibodies suitable for those features.

## Discussion

Here, we report unique high-content datasets that make it possible to assess how ERK activity relates to its downstream effectors at the single-cell level. Our approach, combining live imaging followed by multiplexed immunofluorescence and quantitative models, establishes proof of principle that single-cell ERK activity measurements and endpoint ETG staining contain mutual information and can be modeled bidirectionally, allowing the inference of key parameters in each domain from observations of the other. Our data show that the relationship between activity dynamics and individual cell expression states is significant, despite the inherent noisiness of signal transduction pathways at the single-cell level[33].

Previous modeling efforts in this area have focused on single ETGs[17,19], on multiple genes at the mRNA level[13], or on the population average of ETG proteins[21]. Our analysis offers the advantage of examining multiple ETGs while retaining single-cell relationships, enabling models that account for both cell-to-cell variation and differences between ETGs. In this type of dataset, cell-to-cell variation becomes an asset rather than a limitation, where the thousands of individual cells represent semi-independent trials that can be leveraged to establish complex relationships, even when a relatively small number (~20) of external conditions are provided. Our results quantitatively validate the existing but imprecise concept that certain genes such as c-Fos or Egr-1 represent markers of transient ERK activity, whereas others such as Fra-1 are markers of sustained activity. Our data also place these markers within the context of additional genes such as c-Myc, enabling a more comprehensive model of how ETGs collectively compute responses to dynamic ERK activity.

The approach of combining multiplexed live- and fixed-cell analysis on the same cells is finding increasing utility and has been used to interrogate details of CDK-mediated cell cycle regulation[34], SMAD-regulated stem cell differentiation[35], and the dependence of calcium signaling on gene expression state[36]. The modeling approaches we develop here provide new capabilities for such datasets. We show that combining fixed-cell measurements of these outputs can reliably identify ERK signaling activity history in different cell lines, despite the complication that these relationships vary somewhat between cell lines.

Our models indicate that single-cell ETG measurements provide reliable information about two main types of ERK behavior: long-term history and short-term activation. However, the unexplained variance in ERK activity in our single-cell models raises the question of which other cellular parameters could improve ERK history predictions. Our analysis of simulated ETGs demonstrates that additional ETG measurements could more finely resolve signaling behaviors, especially for features such as Frequency. These targets could be identified by screening transcriptomic or proteomic datasets for their expression responses to ERK[13]. Additional information could also be obtained by combining antibody measurements with multiplexed single-cell transcript measurements, such as MERFISH[37]. Our work also shows that using expression data from groups of cells, rather than individual cells, provides improved prediction of the average ERK activity for the group. Grouping cells in this way makes sense because ERK activity is often important for collective cell behaviors, such as generating patterns of collective migration[38] or regulating cell death and survival in proportion to cell density[39]. Finally, orthogonal markers of internal cell state provide contextual information that improves signaling prediction models[40]. It is feasible that such contextual markers, including protein translation rates, organelle morphology, chromatin state, or cell type identity, could improve predictions.

Several aspects of our analysis suggest that the quantitative relationship between pathway activity and gene expression evolves and may be under selection for certain characteristics. The ETGs that we measured in this study appear to be more biased toward control by the duration of ERK activity (as opposed to amplitude, Fig. 4), than would be expected per the ODE model and simulations (Fig. 7). Such an effect could arise from several factors, including saturation of a particular gene's response to ERK. A key concern is that our ERK biosensor may itself become saturated and fail to capture high ERK activation levels, but we have directly accounted for this measurement issue by calibrating the reporter to provide a linear readout of ERK substrate phosphorylation[41]. A remaining possibility is that the parameters of the ETG response have been selected for sensitivity to duration rather than to amplitude during evolution, which would be consistent with the finding that the relative timing of biochemical events is an important mode in gene regulation[42]. Our data from cancer cells, in which ERK-ETG response functions appear significantly weaker than in non-tumor MCF10A cells, further imply the malleability of these quantitative parameters. It is unclear what drives this change within the micro-evolutionary setting of a tumor, as it appears to occur similarly across the tumor cells examined. The loss of signaling regulation appears to be independent of the driver mutation and may be a late development in tumorigenesis, consistent with models that show initial dependence on driver mutations, followed by dysregulated signaling after various checkpoints are lost[43]. Notably, however, MCF7 cells, which contain a PIK3CA mutation, show the loss of correlation more strongly than lines driven by ERK pathway mutations (A549, HCC827). More work is needed to understand these differences and assess how they are influenced by selective pressures.

In principle, models such as the ones generated here (Fig. 6) could be used to determine important details of ERK signaling dynamics within fixed tissue samples in a clinical setting. For example, the ability to infer the long-term patterns of ERK activity in samples from patients treated with RAS, MEK, EGFR, or other targeted pathway inhibitors could provide a more reliable indication of the effectiveness of long-term ERK activity suppression than current markers, helping to reveal areas of drug resistance. It is also possible that such models could assist in diagnosing the dominant mechanisms of cell signaling within a tumor (or within sub-regions of a tumor), especially in cases where analysis of genetic alterations is ambiguous (e.g., when variants of unknown significance are present). By analogy, single-point measurements of hemoglobin A1C provide a broadly reliable indication of a patient's time-averaged blood sugar that is useful in the clinical management of diabetes.

Our analysis of cancer cell lines adds needed depth to the question of how ERK signaling differs between normal and tumor cells. We find several key parallels between the cell lines examined here and previous biosensor data. The sustained ERK activity found in RAS mutant A549 cells is similar to the signaling phenotypes observed when expressing oncogenic RAS or RAF in non-tumor epithelial cell lines[11] or engineered mouse embryonic fibroblasts[41]. Similarly, the highly sporadic pattern of ERK activity in EGFR mutant HCC827 cells is similar to kinetics found in models of other EGFR- or receptor-driven model systems[11,29]. These parallels strengthen the case that ERK kinetics cluster into similar forms according to the activating mutation in the pathway. Furthermore, we found weaker ERK-ETG relationships, consistent with the degradation of information transmission observed in tumor cells[27]. While this loss of correlation poses some challenges for using the same models across different cell types, our data

nonetheless support the utility of multiplexed immunofluorescence datasets for capturing cell signaling states via histological methods. For example, Fra-1 is consistently correlated with long-term ERK activity, while Egr-1 and pERK correlate on the short term, across all models tested. Combining these stains could potentially distinguish cells with sustained ERK activity (e.g., RAS mutants[41]) or cells with highly stochastic ERK activity (e.g., EGFR-driven cancer cells[11,29]) from cells in which ERK is activated in an organized pattern (e.g., non-tumor tissue[44–46]).

Notably, while all three cancer cell lines examined lack the pRb-Fra-1 correlation seen in non-tumor cells, they retain the pRb-c-Myc correlation. (Fig. S8d). Such changes suggest that c-Myc remains more tightly coupled to proliferation than other ETGs and imply that targetable tumor-specific pathway dependencies can be detected in fixed histological samples. The disruption of c-Jun, Egr-1, and Cyclin-D1 regulation in cells with B-Raf mutations suggests additional such signatures that could be detected[27,47]. Furthermore, the coherence between ETGs within the same cell, available in our datasets but not explicitly explored here, could provide an indicator of transformed signaling activity. Thus, our study supports the general feasibility of inferring functionally significant features of activity dynamics from fixed cell immunofluorescence. A significantly broader training set across tumor samples and primary cells will be needed but would be conceptually straightforward given the analysis framework developed here. Further incorporation of multiplexed signatures of other dynamically regulated pathways such as the cell cycle[48] or metabolic and stress response signaling could also support both generalized and patient-specific models.

Signaling by receptor tyrosine kinase pathways is inherently a multi-input, multi-output system, and here we explore only a limited number of ligand inputs and a restricted set of signals and downstream targets. While serum contains a mixture of ligands, a wider diversity of stimuli needs to be considered to span the varying dynamics that can be driven by receptors including FGFRs, IGFRs, PDGFRs, and TrkA/B[30,41,49]. It will also be important to consider parallel pathways such as PI3K/AKT/mTOR and JAK/STAT and their target genes to fully characterize the regulation of cell state by these receptors, as these pathways work in coordination to determine cellular phenotypes[50,51]. Another key caveat in our datasets is that EKAR3.1, like many ERK biosensors, is partially sensitive to cyclin-dependent kinases (CDKs) during mitosis[52]. Although mitotic events are rarer than ERK activity changes, some variation in EKAR measurements likely arises from CDK activity, and thus some of the correlation between EKAR and pRb is likely attributable to this cross-specificity. This point serves as a reminder that co-variance or cross-talk among measurements will bias these types of machine learning analyses, and should be carefully evaluated. Finally, while datasets on cell lines are sufficient to generate an initial framework, future models will need to be developed by collecting data from primary tissue, patient-derived organoids, or other in vivo systems, to extend these models toward physiological relevance.

## Methods

See Supplementary Table 3 for a list of reagents, materials, and software used in the study.

### Reporter cell line generation

The EKAR3.1 construct was produced from the EKAR3 plasmid by excising the BamHI-MfeI fragment and replacing it with the annealed oligos 5'- gatccgctccagatgtccctagaactccagtggataaagcaaagctgtcattccaa tttccgc and 5'- aattgcggaaattggaatgacagctttgctttatccactggagttctaggg acatctggagcg. This modification introduces an altered substrate sequence with an alanine added between a serine-proline pair near the ERK targeting region, removing a potential secondary phosphorylation site that would be detrimental to the linearity of the reporter

signal relative to ERK activity. Stable cell lines were created by electroporating MCF10A (clone 5e), MCF7, HCC827, or A549 cells with the EKAR3.1 construct on the piggyBAC transposase system[53]. Cells were selected with neomycin (250 µg/ml 2 weeks) until they were resistant to selection (~2 weeks).

### Cell culture and media

MCF10A cells (clone 5e)[54] were maintained in DMEM/F12 supplemented with 5% horse serum, 20 ng/ml EGF, 10 µg/ml Insulin, 500 ng/ml hydrocortisone, and 100 ng/ml cholera toxin. HCC827 (ATCC, CRL-2868) and A549 (ATCC, CCL-185) cells were maintained in RPMI supplemented with 10% Fetal Bovine Serum (FBS) and 2.5 mM L-Glutamine. MCF7 (ATCC, HTB-22) cells were maintained in DMEM supplemented with 10% FBS. 10 cm plates were passaged approximately every four days and re-plated at a 1:10 dilution. Imaging experiments were conducted in custom DMEM/F12 lacking phenol red, riboflavin, and folate. This "imaging media" was supplemented with 500 ng/ml hydrocortisone, 17.5 mM glucose, 1 mM sodium pyruvate, 2 mM glutamine, 50 µg/ml penicillin/streptomycin. Before plating cells for imaging experiments, 5 µl of Rat tail collagen was added to the middle of each well of a glass bottom 96-well plate (Cellvis) and incubated for 45 mins at 37 °C. Cells were trypsinized, plated at 6000 cells per well, and then incubated at 37 °C for 45–60 mins. Growth media was then added, and the plate was incubated overnight. The next day, immediately before the imaging experiment, the plate was washed 3x with imaging media, and the media was changed to imaging media. The experiment began one hour after this media change. In experiments containing cancer cells, HCC827, A549, and MCF7 cells were plated in tandem, with each cell line plated on 1/3rd of the available wells on the plate.

### Live cell microscopy and data acquisition

Prepared 96-well plates were imaged on a Nikon Ti-E inverted microscope with a stage-top incubator (37 °C, 5% CO$_2$). Coordinates within each well of the 96-well plate were imaged at 6 minute increments which were automated by the Nikon Elements AR software. Images were captured using an Andor Zyla 5.5 scMOS camera and a 20x/0.75 NA objective. Chroma #49001 (ET-CFP) and #49003 (ET-YFP) excitation/emission filter cubes were used for mTurquoise2 and YPet measurements, respectively. Further details are described in ref. 53. Coordinates of each acquisition area were saved for future imaging of immunostaining experiments.

### Cyclic immunofluorescence

Immediately after the final acquisition of the live cell experiment, cells were fixed in freshly prepared 12% paraformaldehyde for 10 min. Cells were then permeabilized with fresh, cold methanol for 10 mins (2 times total). Cells were then ready for iterative rounds of staining (4i) using a protocol adapted from ref. 31. Briefly, the iterative protocol involves rounds of elution, blocking, primary staining, secondary staining, Hoechst staining, and finally image acquisition in a specific imaging buffer. Recipes for buffers are as follows: Elution buffer (0.5 M Glycine, 3 M Urea, 3 M Guanidinium Chloride, 70 mM TCEP), Blocking buffer (200 mM NH$_4$Cl, 300 mM Maleimide, 2% BSA in PBS), primary/secondary staining buffer (200 mM NH$_4$Cl, 2% BSA in PBS), Hoechst-33342 stain (1:10,000 in PBS), and 4i imaging buffer (700 mM N-Acetyl Cysteine). Antibodies were incubated for 24 to 48 hours from varying concentrations recommended by the manufacturer. For the MCF10A-only experiments, the protocol was validated during the first replicate experiment to ensure that antibodies were eluted, data is shown in Fig. S2b–d. For the second and third replicate experiments, a visual inspection was completed prior to each round of staining to ensure proper antibody elution. In the cancer cell experiments with additional antibody stains, a visual inspection was conducted between rounds to ensure proper elution.

## Phos-tag western blotting

MCF10A 5e cells were plated on 6-well dishes the day before lysing. Cells were treated with EGF, PD0235901, or imaging media and lysed at the indicated time points. This procedure involved rinsing each well twice with ice-cold PBS, cell scraping, and lysis with RIPA buffer (Sigma) with a Halt protease inhibitor cocktail and 1 μM DTT. Cells were lysed at 80–90% confluency with 50 μl of lysis buffer per well. 2 μl of each sample was then loaded in pre-cast phos-tag gels (Wako-Chem) and ran at 100 V for 3 hours. The gel was chelated two times with transfer buffer and 10 mM EDTA for 15 minutes each and rinsed once more with just transfer buffer. Proteins were transferred overnight at 50 V. The membrane was blocked with Li-COR Odyssey blocking buffer and blotted with anti-GFP antibody (24 hr incubation). The membrane was then blotted with Li-COR 800 anti-Mouse secondary antibody and imaged using a fluorescent scanner (Sapphire-Azure Biosystems). Intensities of the resulting phosphorylated EKAR3.1 reporter and total EKAR3.1 bands were measured in ImageJ.

## Image processing

Imaging data were saved as .nd2 files and accessed using the Bio-Formats toolbox for MATLAB (available from www.openmicroscopy. org/bio-formats), and processed with a custom MATLAB cell segmentation pipeline[53]. The procedure identified each cell's nucleus using either EKAR3.1 (live-cell) or Hoechst 33342 (IF) as a nuclear marker. The cytoplasm was defined as a ring around each cell's nucleus. Background signal intensity was measured by imaging a well with no cells, but containing live-cell imaging media or 4i imaging buffer. Cell position tracking and linking were performed using uTrack 2.0[55]. The resulting single-cell data was filtered to remove cells with less than 15 hours of tracking data. FRET measurements of ERK activity for each cell were calculated with $1 - ((CFP/YFP) / R_p)$, where CFP and YFP are the intensities of Cyan and YFP channels measured in each cell, respectively. $R_p$ is the ratio of total power collected of CFP over YFP where the power of each channel is the integral of the spectral product of excitation intensity, filter transmittances, exposure time, fluorophore absorption and emission properties, and quantum efficiency of the camera (detailed in appendix of ref. [41]). To link live-cell and fixed-cell data together, X-Y coordinates of cells in each well were registered to align cells across measurements. For wells with troublesome alignment results, image registration was conducted to calculate the shift in orientation between the images.

## Batch effect correction

To correct for batch effects in the immunofluorescence measurements across three replicates of MCF10A experiments, we scaled measurements of each target to optimally align median values across identical treatments. As many measurement distributions were sufficiently long-tailed to bias even median values, the normalization was performed in logspace. For each 4i target, we calculated the median value for each treatment and matched identical treatments across replicates. These treatments included all EGF doses at timepoint 30, MEKi at timepoint 30, and imaging media control. We then took the $\log_{10}$ of these values and fit a linear model (Eq. 1):

$$Intensity_{replicate3} = \beta_1 \left( Intensity_{replicaten} \right) + \beta_0 \qquad (1)$$

Where $Intensity_{replicate\_n}$ represents $\log_{10}$ median values for either replicate 1 or replicate 2, $Intensity_{replicate3}$ represents the corresponding $\log_{10}$ median values for the third experimental replicate, and $\beta_0$ and $\beta_1$ are the scaling factors. These scaling factors were then used to correct all single-cell values for replicates 1 and 2. The corrected values were then returned to the linear scale by exponentiating. The normalization resulted in effective batch correction across all conditions. (Fig. S2c, d). A similar method was conducted in cancer cell experiments; however, since the three cell lines were plated in tandem and our goals was to

preserve cell-line differences, we computed normalization samples as the median over all cell lines for each condition in each replicate. As such, all cell lines in each replicate were scaled with the identical scaling factor($\beta_1$) per antibody. A visualization of these normalization results for each cell line is shown in Fig. S5d, S6d, S7d. Nuclear and Cytosolic measurements for each antibody stain were normalized individually.

## EKAR3.1 Calibration

FRET measurements from MCF10A cells were calibrated to deliver a quantitative linear readout of ERK activity, as described previously[41]; refer to the Appendix of this past study for a detailed derivation of this method. Briefly, we used Phos-Tag immunoblotting to quantify the fraction of the EKAR3.1 reporter that is phosphorylated in 3 concentrations of EGF (15 mins), phosphorylation inhibited (MEKi for 2 hours), and control conditions (Supplementary Figs. S1b and S11). These values were then linearly fit against the average EKAR3.1 signal for the same conditions (Eq. 2).

$$\frac{EKAR^A}{EKAR^T} = (K_{AU} + K_{AP}) \frac{EKAR^P}{EKAR^T} \qquad (2)$$

Here, $EKAR^P/EKAR^T$ represents the phos-tag ratio between phosphorylated and total reporter. The EKAR3.1 signal is pre-processed to estimate the fraction of EKAR molecules in an "associated" conformation ($f_A$ or $EKAR^A/EKAR^T$), i.e.: $f_A = \frac{EKAR^A}{EKAR^T} = 1 - \frac{I_{CFP}}{I_{YFP}} / R_P$, where $I_{CFP}$ and $I_{YFP}$ refer to the measured intensity of the cyan (donor) and yellow (acceptor) channels, respectively, and $R_P$ refers to the corresponding ratio of imaging power in each of these channels. $K_{AU}$ and $K_{AP}$ represent the fractions of EKAR in the "associated" state when completely unphosphorylated and when completely phosphorylated, respectively. Single-cell FRET measurements (i.e. $f_A$) were then used to estimate the concentration ratio of active ERK to the competing phosphatase(s) (Eq. 3). This ratio is the quantitative measure of ERK activity in a cell ($ERK^A/PPASE^A$).

$$\frac{ERK^A}{PPASE^A} = \frac{f_A - K_{AU}}{K_{AP} - f_A} \qquad (3)$$

## Data analysis and regression modeling

Cells with less than 15 hours of data were removed prior to analysis, and cells out of the expected range of the FRET measurements were removed. FRET measurements were then adjusted using the reporter calibration model created from the phos-tag experiments. Thus, statistical models were created on cells that had complete EKAR and ETG measurements. Models were created using 10-fold cross-validation. The data were randomly assigned to 10 groups, with the $10^{th}$ group held out of the model fitting procedure. The model was then tested against the $10^{th}$ group (test-set) to collect the test error (residual mean squared error, RMSE). This procedure is repeated for a total of 10 times to collect RMSE values from 10 test sets. The model that produced the lower test-set error was then refitted to the entire dataset to calculate the reported RMSE values.

## Pulse analysis and peak detection

The findpeaks function in MATLAB was used to find local maxima (peaks) for each cell's ERK activity. Pulse features were then calculated based on the identified peaks. Frequency was calculated using the meanfreq function in MATLAB. This function estimates the mean normalized frequency of the power spectrum of each ERK activity trace.

## Statistical tests

For single-cell immunofluorescence data, each statistical comparison was made by t-test with unequal variances, and false discovery rate was

controlled within each dataset via the Benjamini and Hochberg Step-Up procedure ($\alpha = 0.05$). The variance for each experiment was determined from single-cell samples and added to the variance across experiments. This corresponds to a linear error model: $\varepsilon_i = \varepsilon_{cell} + \varepsilon_{exp}$, where the error (from the mean) of an individual cell $\varepsilon_i$ equals the sum of the errors arising from cell-to-cell variation $\varepsilon_{cell}$ and from experiment variation $\varepsilon_{exp}$.

## Spatial heatmap generation

Each cell's time-averaged coordinates were used to calculate the average Euclidean distance between each pair of cells within each well of the 96-well plate. Hierarchical clustering was performed on this distance matrix. The optimal leaf order was calculated by maximizing the sum of the similarity between adjacent leaves by flipping tree branches and without dividing the clusters. This order was then used to sort and display the live-cell and fixed-cell data.

## ETG prediction models and evaluation

CNN models consisted of 1) a feature learning module and 2) a prediction module. The feature learning module consists of 2 convolutional layers (16 channels and kernel size of 16) followed by an FC layer with a size of 192 to match the initial input size. The prediction module consists of 2 FC layers (each size of 64 with relu activations) followed by a final linear FC layer that outputs a single ETG prediction. We performed hyperparameter variation for the learning rate ([0.01,0.001,0.0001]) and L2-regularization ([0.1,0.01,0.001]) to compare the average test prediction performance (MSE). While the performance was robust across parameters, we chose a learning rate of 0.001 and L2-regularization of 0.01, which had the best overall performance. We trained one model per ETG for 100 epochs using the Adam optimizer with a learning rate of 0.001 and L2 regularization of 0.001. For the linear model, linear regression was implemented using the sklearn python package with default parameters. The inputs were either the raw or featurized ERK activity for the linear model or the featurized linear model, respectively. Evaluation of the model was performed by holding out 20% of the total data as a test set per fold, then splitting the remaining data between the training set, consisting of 64% of the total data, and the validation set consisting of 16% of total data,.with each fold having roughly the same representation from each well of origin and treatment.

To identify significant time points, we used feature attribution, specifically Integrated Gradient[56], to identify input time points that the model considers significant to the prediction of ETG. Integrated Gradient was implemented using the Python package Captum[57]. Feature attribution outputs score from each input time point to ETG per cell, which was averaged across cells for summarized visualization in the form of a heatmap.

To test the importance of the timepoints after the initial stimulation, we trained new CNN models to only use timepoints 2 hours after stimulation for ETG prediction. This model was trained on 14 hours of ERK activity data. The model and training setup used is identical to the setup used for the model with all the time points (19 hours of ERK activity data).

While recurrent neural networks (RNNs) or sequence-based models are typically better suited for time series data, they are more difficult to train, especially for noisy continuous data. For our simple regression task with the additional goal of identifying the most informative time points, CNN was sufficient. Additionally, RNNs would have a harder time modeling spikes from stimulus as it cannot consider absolute time/position. While more advanced sequenced models such as attention-based models could have been used, the goal here was not necessarily to achieve the best prediction possible but to identify significant time points. Therefore, we chose to use CNN for our analysis with MSE and $R^2$ metrics for evaluation.

## Temporal ERK Signaling Classifier

EKAR time series data were clustered into five groups using k-means clustering. Each group was assigned its class label. Because the k-means algorithm clusters all cells into a respective class, individual cells whose signal did not correlate with the mean of the class were removed. To do so, we calculated the Pearson correlation ($r$) between each cell and the mean of its class and enforced a minimum threshold of $r = 0.7$. Next, an ensemble learning method (AdaBoost M2[58] with 500 learning cycles) was fit using antibody staining measurements to predict the class labels of each cell. This was done using MATLAB's fitcensemble function. 10-fold cross-validation was used and the model with the lowest test set error was displayed. Predictor Importance was estimated by summing the estimates over all weak learners in the ensemble–MATLAB predictorImportance(*ens*) function. Regional and Whole-well models were trained similarly, except that the data was first grouped into regional neighborhoods or well averages. For whole-well models (Fig. 6e, green bars), a standard decision tree classifier was used due to the small sample size. MCF10A classification models used nuclear pERK, Fra-1, c-Myc, c-Jun, (p)c-Fos, Egr-1, and pRb as predictors, whereas cancer cell lines additionally used $DUSP1^{nuc}$, $DUSP6^{cyt}$, $p4EBP1^{nuc}$, $FoxO1^{nuc}$, $RSK1^{nuc}$, $GSK3\beta^{nuc}$, $EZH2^{nuc}$, $pEGFR^{cyt}$, $pp70S6k^{cyt}$, $NF\text{-}1^{cyt}$, and E-Cadherin$^{cyt}$.

## Ordinary Differential Equation Modeling

The ODE model was adapted from Davies et al.[29]. The model of ERK-dependent gene expression (Eqs. 4–7) was constructed from a mass action approximation. This process is modeled in four steps (Eq. 4) phosphorylation of a transcription factor by ERK ($TF^P$), (Eq. 5) transcription of target mRNA (mRNA), (Eq. 6) translation of target protein (P), and (Eq. 7) potential stabilization of target protein by ERK-dependent phosphorylation ($P^P$). A regulatory term is included in the transcription process allowing negative feedback from the target protein onto its own production. The model is formulated as a delay differential equation to account for the effective lag times of transcription and translation without explicitly addressing the complex processes involved. For the purpose of simulating a set of specific hypothetical ETGs, we selected a reduced parameter space from which to sample (see Supplemental Table 2). Given the relative simplicity of the model, the parameters to vary were chosen analytically, avoiding direct correlations and minimizing the sampling space dimension.

$$\frac{d}{dt} TF^P(t) = k_{pTF} * ERK(t) * \left( TF^T - TF^P(t) \right) - k_{dTF} * TF^P(t) \quad (4)$$

$$\frac{d}{dt} mRNA(t) = \frac{k_b + k_m * TF^P(t - \tau_m)}{\left( \frac{P(t-\tau_m) + P^P(t-\tau_m)}{k_D} \right)^v + 1} - k_{\varnothing m} * mRNA(t) \quad (5)$$

$$\frac{d}{dt} P(t) = k_P * mRNA(t - \tau_P) + k_{dP} * P^P(t) - (k_{\varnothing P} + k_{P^P}) * P(t) \quad (6)$$

$$\frac{d}{dt} P^P(t) = k_{pP} * ERK(t) * P(t) - \left( k_{\varnothing P^P} + k_{dP} \right) * P^P(t) \quad (7)$$

## UMAP projections

To embed the high-dimensional immunofluorecence data into 2 dimensions for visualization, we used the UMAP algorithm[59]. UMAP was run using the umap-learn python package. Default hyperparameters were used: n_neighbors = 15, min_dist = 0.1, and n_components = 2. UMAPs of MCF10A data were created with nuclear measurements of Egr-1, Fra-1, c-Jun, c-Myc, c-Fos, pERK, pc-Fos, and pRb. UMAPs of

HCC827, A549, and MCF7 were created with nuclear measurements of Egr-1, Fra-1, c-Jun, c-Myc, c-Fos, pERK, pc-Fos, pRb, DUSP1, p4EBP1, FoxO1, RSK1, GSK3β, EZH2, along with cytosolic measurements of pEGFR, pp-70s6k, NF-1, E-Cadherin, and DUSP6.

## Reporting summary
Further information on research design is available in the Nature Portfolio Reporting Summary linked to this article.

## Data availability
All source data for figures, including all processed FRET and immunofluorescence values for single cells, have been deposited in a Figshare repository and are available at this link: https://doi.org/10.6084/m9.figshare.27047617. Due to the file size limitations for existing repositories, raw microscopy image sets will be provided by the corresponding author upon e-mailed request.

## Code availability
All custom MATLAB and Python code for model generation, data analysis, and figure plotting are provided here: https://doi.org/10.6084/m9.figshare.27047617.

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

## Acknowledgements

This work was supported by the National Institute of General Medical Sciences (R01GM115650 and R35GM139621 to JGA; T32GM007377 and 5R25GM056765 to AR) the National Heart, Lung, and Blood Institute (R01HL151983 to JGA), and the National Cancer Institute (U54CA283766). We thank Taryn Gillies for their helpful feedback on the manuscript.

## Author contributions

A.R., M.P., and J.G.A. conceptualized the study, interpreted data, performed data analysis, and wrote the manuscript. A.R. and C.T. conducted imaging experiments. M.P. created the ordinary differential equation model and the scripts for gene simulations. Y.C. and G.Q. performed convolutional neural network analysis. D.M. assisted with immunoblots and image registration. M.C. assisted with cell culture. N.K. assisted with data analysis.

## Competing interests

John Albeck has received research grants from Kirin Corporation. The other authors declare no competing interests.

## Inclusion and Ethics

All collaborators have fulfilled the criteria for authorship required by Nature Portfolio and are included as authors of this study. This research is locally relevant and included local researchers throughout the entire research process including study design, study implementation, data ownership, intellectual property, and authorship of publications. Roles and responsibilities were agreed amongst collaborators ahead of the research. This research was not severely restricted or prohibited in the researchers' setting. This work does not result in stigmatization, incrimination, discrimination or otherwise personal risk to participants, nor does it risk the health, safety, and security of the researchers. We have considered relevant local and regional research in the citations.
