## [Transparent Peer Review file · Nature Communications]

Deciphering the History of ERK Activity from Fixed-Cell Immunofluorescence Measurements

Corresponding Author: Professor John Albeck

Version 0:

Reviewer comments:

Reviewer #1

(Remarks to the Author)

This is a very interesting study. The authors have adopted multiple modeling approaches to model ERK signaling, a very important biological process. However, none of the models really demonstrated very high performance on prediction based on the R^2 values. The analysis methods adopted were common methods. There are some defects in the analysis methods. Due to the low R^2 score, a quantitative conclusion is hard to draw.

Major Concerns:

- 1) Since there are cells from each of the 96 wells, a UMAP might be adopted to visualize the batch effect correction to have a good quality control of the data.
- 2) For Figures 5a and 5d, do authors interpret the CNN models with the best performance or average performance of multiple data partitions? What's the activation used in the CNN model? More hyperparameters should be included for verification. How many data partitions have been used? What's the overall size of the data for training, testing, and validation? Providing this information is very important to reproduce the results.
- 3) Interpretation of NN models is not only dependent on weights but also on the value of the nodes and activation functions. In addition, for time series, the RNN model is normally chosen, what's the reason for using CNN for time series data? Especially, the CNN models were not really validated without an evaluation based on the confusion matrix or precision and recall.
- 4) For the ODE models, the sensitivity of parameters should be performed to select the most sensitive genes as model variables. In addition, the selected 1,000 genes should consider the convergence time. The fast convergent genes is not as important as the genes that respond slowly. Or different time scales should be considered for fast and slow responses separately. Range of parameters should be provided as a supplementary document.

Minor Concerns:

- 1) Figure 1C should be further explained for readers to understand the biological meaning of the measurement. Figure 1d demonstrated two responses with the same EGF 10ng/ml at different times. The red vertical lines were shown without any explanation.
- 2) On lines 154 and 155, an R^2 score below 0.42 does not represent a good prediction. Therefore, it's not sound to claim "These results suggest the duration of ERK activation seems to have a stronger influence on gene expression than the strength of the activation."
- 3) What's the "All" case in Figure 3e. The R^2 in "All" case is the highest with respect to each ERK parameter.
- 4) The styles of the fonts were not consistent.

Reviewer #2

(Remarks to the Author)

Ram et al. investigate the ERK signaling dynamics using a live, single-cell ERK biosensor approach along with multiplexed immunofluorescence staining of downstream target proteins.

Additionally, combining linear regression, machine learning, and differential equation models they develop an interpretive

framework for combination of immunostainings. In the case of these particular epithelial cells, long term activation of ERK signaling correlates with Fra-1 and pRb levels, while Egr-1 and c-Myc indicate recent activation. The work attempts to address heterogeneity within the ERK dynamics in a population of cells, with the long-term aim of providing a tool for effectively annotating ERK dynamics within fixed tissues.

Major points:

1. The biological significance of the data.

There are questions, as the authors state, on the specificity of the live reporter; and the observation are based on one single epithelial cell line in 2-D cultures. It is dubious whether they would apply to normal or tumor tissues.

An experimental design including more cell types/cell lines and in the case of tumor cells spheroids or even patient-derived organoids would have been more informative and biologically relevant.

Additionally, it would have been important to test whether different stimuli generate a different ERK dynamics and whether this correlates with similar, or distinct, ETG outcomes. This aspect is important to consider if the authors think of transposing their model to tissue, which are exposed to a plethora of different stimuli.

Even if we want to stay within this limited experimental design, have the authors tested the model to show whether predicted signaling histories actually match the behavior of the live reporter?

Answers to such questions would be needed for a generalist journals such as Nature Communications

2. Can the model be applied to fixed tissues as the authors claim?

The main difficulty in assessing whether the models derived from the data in the manuscript apply to fixed tissue samples is the lack of the possibility of testing the model *in vivo*.

3. Even if we hypothesize that the model applies to tissues, the value of inferring the dynamics of ERK signaling in tissues is not made clear to the reader.

Minor points:

1. More information on why these particular 8 target genes were selected would have been helpful.

2. Related, if the model should be applied to complex tissue samples, the question of which ETG to use and which ETGs would be expressed and with which dynamics has not been addressed, even speculatively.

3. Fig 5 A and B- cluster 4 and 5 appear mislabeled.

Reviewer #3

(Remarks to the Author)

In this study, Ram et al. investigated whether the history of ERK activity dynamics can be predicted from the expression levels of ERK target genes by analyzing ERK activity dynamics with FRET imaging and cyclic immunofluorescence. First, the authors added different concentrations of EGF and/or MEKi to EKAR3.5-expressing MCF-10A cells, fixed the cells at the endpoint, and obtained expression levels of pERK, Fra-1, c-Myc, and others using 8 different antibodies (Fig. 1). Using these large amounts of data, they investigated the relationship between ERK dynamics and ETGs by correlation (Fig. 2) and regression analysis (Fig. 3). Furthermore, using CNNs, they predicted the expression of ETGs based on the dynamics of ERK activity (Fig. 4), predicted the dynamics of ERK activity based on the levels of ETGs using a decision tree model (Fig. 5), and analyzed the potential dynamics of ETGs using an ERK-dependent gene expression model (Fig. 6). These analyses show that Fra-1 and pRB levels reflect the duration of ERK activity, while Egr-1 and c-Myc indicate the recent level of ERK activity. The authors also suggest the possibility of predicting ERK activity dynamics from multidimensional immunostaining in the future.

Overall, the work is, for the most part, sound, and the experiments are well-designed. I really enjoyed reading this excellent piece. I provide some comments below that would be worth addressing in a revised version.

1. The data presented here are mainly analyzed for the dynamics of ERK activity and correlation analysis with ETGs. However, the data also include spatial information, but with the exception of Figs. 2d, 2f, and 5e, the spatial information is mostly ignored. In fact, several groups have reported that ERK activity propagates to neighboring cells. Can the authors show whether they can predict ETGs more accurately by using the ERK activity dynamics of neighboring cells, and whether they can predict ERK activity dynamics more accurately by using the information on the ETGs of neighboring cells? The spatial transcriptome is currently available, and I thought that it would make the significance of this paper stand out even more.

2. Line 99, Figure S1a-d: The authors should clarify the mode of action and meaning of calibration of EKAR3.5. First, it should be stated in the text that EKAR3.5, like other ERK biosensors, measures the balance between phosphorylation by ERK and dephosphorylation by phosphatase. Second, there is no rationale for the FRET measurement to be linear with respect to the phosphorylation level of EKAR3.5 in Figure S1d. Perhaps it should be fitted to the Hill function, in which case more data points would be needed, but I am not going to ask the authors to do this. Further, if it was a linear fit, then the calibration essentially has no effect on the later analysis at all. Essentially, FRET measurements should be correlated with pERK levels, but this is also outside the scope of this paper. In any case, please add an explanation of the mode of action of EKAR3.5 and the meanings of calibrations so that readers do not misunderstand.

3. I presume that the response function of ERK activity dynamics for each ETGs by linear regression model would further explain clearly the authors' claim that Fra-1, pRB are induced by sustained ERK activity and EGR-1, c-Myc by transient ERK activity.

4. Line 280: I did not understand what the authors wanted to show in Figure 5e. Did they classify ERK activity dynamics from immunostaining data and map it to cells? Or did they acquire new data and validate it? Could you please add some more explanation?

Version 1:

Reviewer comments:

Reviewer #1

(Remarks to the Author)

The authors responded well to my questions in the previous submission, alleviating my concerns about the R2 for prediction accuracy.

I have not found the source codes related to the manuscript for further verification. Specifically, multiple programming languages and algorithms have been adopted. Sharing instructions on data processing with source codes will help readers better understand the results.

(Remarks on code availability)

No code is uploaded.

Reviewer #3

(Remarks to the Author)

The authors have done a mostly adequate job of responding to the reviewers' comments.

(Remarks on code availability)

Reviewer #4

(Remarks to the Author)

This timely study characterizes the ERK signaling dynamics associated with specific and distinct effector outputs. It is a resource for the signaling community and timely in terms of shedding light on which effectors might best be used clinically to test for residual ERK activity. The work is quite comprehensive, but could still be improved by correlating the effector output with specific cell phenotypes. Specific comments are below.

1. The interpretation of the EGF treatments (different concentrations and durations of treatment) in Figure S3 is limited by the focus on only EGF. In vivo, cells are exposed to many different growth factors at the same time and rarely experience a complete starve followed by a single growth factor stimulation. The authors try to address this more physiological approach by including stimulations in the presence of serum in figures S5-7. While this does not completely address the question of how translatable the current data and model are to cells in vivo, it is a good first step towards what will likely be a future long-term effort to develop this approach.

2. The study in 3 different cancer cell lines is very interesting and key to the generalizability and utility of the conclusions. The discussion and overall conclusions from this expanded study should be improved. The genetic drivers that affect ERK and other key cell proliferation, survival, and migration pathways should be more clearly called out (HCC827 – EGFR, A549 – KRAS, MCF7 – PI3KCA) and put in context with how ERK is known to signal in the presence of these mutations, work by this lab and others. The lack of ETG-ERK activity correlations in the MCF7 line may be due to its dependence on PI3K/AKT signaling, rather than ERK.

3. The conclusions from this extensive 4 cell line, multiple agonist study should be better summarized for interpretation of the biological significance of key findings through a few possible approaches:

- a. The UMAP approach in Figure S5-7 could be applied so that different colors show the individual ETGs, rather than the replicates. If they current presentation, they provide little new information. Do feature populations fall out that explain the different correlations with respect to the cells' driver mutations and dependencies.
- b. Since the relevant ERK signaling features appear to be different in the different cell lines, it is important to determine if the ERK signaling and ETGs are also different for the cell phenotype. Does inhibiting ERK have a bigger effect on the proliferation of HCC827 and A549 cells and a relatively small effect on MCF7 cells? Is this dependent on Fra-1 and/or myc? Does c-Jun correlate better with ERK signaling in A549 cells because the other pathways controlling c-Jun are not active in these cells? What is the meaning of Egr-1 correlation with ERK pulse frequency? Does this suggest an upcoming mitosis, while Fra-1 controls a baseline level of mitotic capability? Understanding what the key features are that suggest meaningful ERK activity will be important for using these readouts in assessing pathway targeting in cancer treatments.

(Remarks on code availability)

Response to Reviewers

Deciphering the History of ERK Activity from Fixed-Cell Immunofluorescence Measurements

Abhineet Ram, Michael Pargett, Yongin Choi, Devan Murphy, Carolyn Teragawa, Markhus Cabel, Nont Kosaisawe, Gerald Quon, John G. Albeck*

Reviewer #1 (Remarks to the Author):

This is a very interesting study. The authors have adopted multiple modeling approaches to model ERK signaling, a very important biological process. However, none of the models really demonstrated very high performance on prediction based on the R^2 values. The analysis methods adopted were common methods. There are some defects in the analysis methods. Due to the low R^2 score, a quantitative conclusion is hard to draw.

We thank the reviewer for appreciating the positive points of our study, and for the helpful comments. Regarding the R^2 values and the performance of predictions, we would like to note that the R^2 values from our models are consistent with other models of biological processes at the single-cell level. For example, Popovic et al. (2018, PMID: 30342881) examined the correlations between mRNA transcript copy numbers and corresponding protein expression levels for 23 genes, and they observed correlation values of $r=0.45-0.86$. R^2 values for models of protein expression based on mRNA ranged from 0.397 to 0.672, depending on the cell cycle phase (see Popovic, 2018, Fig. 2G). Given that mRNA abundance is one of the main factors determining protein abundance for many genes, it seems reasonable to expect that such values represent a best-case scenario for predicting a single-cell protein expression value (or a related value) from molecular measurements made in the same cells. We hope that considering these previously reported values helps to put the R^2 values reported in our work into context, and we have included a short discussion of this point in the Discussion section.

Major Concerns:

1) Since there are cells from each of the 96 wells, a UMAP might be adopted to visualize the batch effect correction to have a good quality control of the data.

We agree that this is a useful visualization. We have now generated UMAPs to visualize the batch effect correction, they are shown as Fig. S2c. We have also included box plots of each stain per treatment, colored by replicate plate, in Fig. S2a.

2) For Figures 5a and 5d, do authors interpret the CNN models with the best performance or average performance of multiple data partitions?

Both the individual and mean validation set results for 5 folds are now presented and annotated in Fig. 3b (bars and gray dots, respectively). We refer to the mean performance of these five

folds in the text referencing this plot. Fig. 3c and d each represent the predicted vs observed values of the validation set from one of the 5-fold partitions (randomly chosen). These results (Fig. 3c,d) were very similar across all of the partitions.

What's the activation used in the CNN model? More hyperparameters should be included for verification. How many data partitions have been used?

We have included these important details in the corresponding Methods section, which now reads as follows:

“CNN models consisted of 1) a feature learning module and 2) a prediction module. The feature learning module consists of 2 convolutional layers (16 channels and kernel size of 16) followed by an FC layer with size of 192 to match the initial input size. The prediction module consists of 2 FC layers (each size of 64 with relu activations) followed by a final linear FC layer that outputs a single ETG prediction. We performed hyperparameter variation for the learning rate ([0.01,0.001,0.0001]) and L2-regularization ([0.1,0.01,0.001]) to compare the average test prediction performance (MSE). While the performance was robust across parameters, we chose a learning rate of 0.001 and L2-regularization of 0.01, which had the best overall performance. We trained one model per ETG for 100 epochs using the Adam optimizer with learning rate of 0.001 and L2 regularization of 0.001.”

Please see the response immediately below regarding data partitions.

What's the overall size of the data for training, testing, and validation? Providing this information is very important to reproduce the results.

We have included these details in the corresponding Methods section, which now reads:

“5-fold cross-validation was performed by holding out 20% of the total data as testing set (14,292 cells) per fold, then splitting the rest between the training set, consisting of 64% of total data (45,233 cells), and validation set, consisting of 16% of total data (11,344 cells).”

3) Interpretation of NN models is not only dependent on weights but also on the value of the nodes and activation functions. In addition, for time series, the RNN model is normally chosen, what's the reason for using CNN for time series data? Especially, the CNN models were not really validated without an evaluation based on the confusion matrix or precision and recall.

While RNN or sequence-based models are specifically designed for time series data, they are not necessary when the variable length and time points match across the entire dataset (as is the case here), at which point the dataset behaves similarly to a one-dimensional image. Additionally, RNNs would have a harder time modeling spikes from stimulus as it cannot consider absolute time/position. Thus, for our regression task, with the additional goal of identifying the most informative time points, CNN was sufficient. While more advanced sequenced models such as attention-based models could technically have been used, the goal here was not necessarily to achieve the best prediction possible but identify significant time points. Therefore, we chose to use CNN for our analysis.

Since we used our CNN model for regression rather than classification, MSE and R^2 are more relevant metrics than precision and recall, which is why we used them for evaluation.

To make these points clear in the paper, we have included these points in the relevant Methods section (“ETG prediction models and evaluation”)

4) For the ODE models, the sensitivity of parameters should be performed to select the most sensitive genes as model variables. In addition, the selected 1,000 genes should consider the convergence time. The fast convergent genes is not as important as the genes that respond slowly. Or different time scales should be considered for fast and slow responses separately. Range of parameters should be provided as a supplementary document.

We agree with this point and apologize that our parameter ranges were not made clear in the initial manuscript. Because the model is relatively small, we were able to explore a parameter space that effectively covered the full range of biologically rational rates for the modeled processes (such as the decay rates of the different states involved). The parameter screen for simETGs was verified to deliver genes that (at least) span the range of response times observed for real ETGs.

We now include Supplementary Table 2, which lists parameters and the sampling range for each, as applicable, and have added the following to the description of the ODE modeling in the Methods.

“For the purpose of simulating a set of specific hypothetical ETGs, we selected a reduced parameter space from which to sample (see Supplemental Table 2). Given the relative simplicity of the model, the parameters to vary were chosen analytically, avoiding direct correlations and minimizing the sampling space dimension.”

Minor Concerns:

1) Figure 1C should be further explained for readers to understand the biological meaning of the measurement. Figure 1d demonstrated two responses with the same EGF 10ng/ml at different times. The red vertical lines were shown without any explanation.

We have updated the legends for Figure 1c and 1d to read as follows:

c Condition average response measurements depicted as a heatmap. Each row represents the average EKAR FRET measurement for a condition, indicated by the color scale (yellow, high ERK; blue, low ERK). EGF concentration is indicated by colored triangles as in Fig. 1b. MEKi = MEK inhibitor PD0325901 (100nM) ($n_{\text{well replicates}} = 2-4$ for each treatment).

d Single-cell response plots for five representative cells in each indicated condition. Bold lines indicate the mean of all cells in one well of the condition. Red lines indicate the time points where treatments were added.

2) On lines 154 and 155, an R^2 score below 0.42 does not represent a good prediction. Therefore, it's not sound to claim "These results suggest the duration of ERK activation seems to have a stronger influence on gene expression than the strength of the activation."

We have modified the section in question to no longer make that claim. The relevant section now reads:

"The top single ETG predictor based on variance explained (R^2) was Fra-1 for Sum of ERK duration ($R^2 = 0.42$). MLR models improved the predictions for all features, with Sum of duration still the best-predicted feature ($R^2 = 0.53$). In comparison, amplitude characteristics of ERK activity, like Max or Average peak height, and features related to pulsatile ERK behavior, such as Average duration and Frequency, were poorly predicted by individual parameters ($R^2 < 0.15$) or by MLR ($R^2 < 0.2$)."

3) What's the "All" case in Figure 3e. The R^2 in "All" case is the highest with respect to each ERK parameter.

The "All" case represents a multiple linear regression using all 8 ETGs.

We clarify this by adding the following text in the Results section (lines 206-207):

"we generated multiple linear regression (MLR) models using all 4i measurements as predictors at once (Fig. 4a-bottom row)".

And by including the following text in the figure legend:

"All" indicates multiple regression models using all ETGs as predictors.'

4) The styles of the fonts were not consistent.

We have updated all figures using a consistent font style throughout.

Reviewer #2 (Remarks to the Author):

Ram et al. investigate the ERK signaling dynamics using a live, single-cell ERK biosensor approach along with multiplexed immunofluorescence staining of downstream target proteins. Additionally, combining linear regression, machine learning, and differential equation models they develop an interpretive framework for combination of immunostainings. In the case of these particular epithelial cells, long term activation of ERK signaling correlates with Fra-1 and pRb levels, while Egr-1 and c-Myc indicate recent activation.

The work attempts to address heterogeneity within the ERK dynamics in a population of cells, with the long-term aim of providing a tool for effectively annotating ERK dynamics within fixed tissues.

Major points:

1. The biological significance of the data.

There are questions, as the authors state, on the specificity of the live reporter; and the observation are based on one single epithelial cell line in 2-D cultures. It is dubious whether they would apply to normal or tumor tissues.

An experimental design including more cell types/cell lines and in the case of tumor cells spheroids or even patient-derived organoids would have been more informative and biologically relevant.

We thank the reviewer for their helpful comments and critiques of our study.

We have now included an extensive dataset including 3 additional cell lines of tumor origin (HCC827 and A549 lung cancer cells, and MCF7 breast cancer cells). We observe distinct reporter kinetics in these cells, providing a wider space of behaviors and increasing the likelihood that cells within tumors or tissues can be compared to the behaviors within our dataset.

Additionally, it would have been important to test whether different stimuli generate a different ERK dynamics and whether this correlates with similar, or distinct, ETG outcomes. This aspect is important to consider if the authors think of transposing their model to tissue, which are exposed to a plethora of different stimuli.

In the expanded dataset, we have included conditions containing fetal bovine serum as a stimulating factor. FBS contains a complex mixture of growth factors and thus reasonably replicates situations in which cells are exposed to milieu of different stimuli converging on ERK activity.

Even if we want to stay within this limited experimental design, have the authors tested the model to show whether predicted signaling histories actually match the behavior of the live reporter? Answers to such questions would be needed for a generalist journals such as Nature Communications

In considering the wider dataset now collected, we realized that focusing simply on predicting kinetics based on IF stains is too narrow of a concept. Our new data demonstrate that the connection of ERK activity to the proteomic changes that it drives is complex and variable. We think that the paper now serves as an important guide to understanding this relationship through quantitative models, in addition to providing what is by far the most detailed single-cell dataset addressing this question. We have therefore reframed our introduction and discussion of the paper significantly, with less emphasis on simply predicting the ERK activity history, and more emphasis on being able to model the relationship between ERK and its targets.

With regard to how well the histories can be predicted, the best visualization is found in the confusion matrix in Fig. 6c, S9c,g,k. This analysis shows the prediction accuracy of each class of activity history, within a given cell line, which substantially exceeds random chance in all cases, although some classes are more accurately predicted than others.

2. Can the model be applied to fixed tissues as the authors claim?

The main difficulty in assessing whether the models derived from the data in the manuscript apply to fixed tissue samples is the lack of the possibility of testing the model *in vivo*.

We thank the reviewer for this comment, and we agree that this is an important question. Our view is that our study provides proof of principle that such predictions can be made, to the extent achievable within an *in vitro* system. Fully validating such predictions within an actual tissue setting would require live and fixed imaging of ERK activity within a tissue. While such imaging has been achieved in a small number of reports, setting up such a system and additionally performing the fixed cell analysis for a sufficiently large sample of cells would be extremely challenging and well outside the scope of what is possible in the current study. We look forward to following up on this question in future work.

3. Even if we hypothesize that the model applies to tissues, the value of inferring the dynamics of ERK signaling in tissues is not made clear to the reader.

We have rewritten the introduction to help make this point more clear. Specifically, the third paragraph of the introduction now directly addresses the possible uses for models that relate ETG patterns and ERK activity.

Minor points:

1. More information on why these particular 8 target genes were selected would have been helpful.

We have updated the last paragraph of the introduction to explain how this choice was made. It now reads:

“To develop such an interpretive framework, we used a live-cell biosensor of ERK activity in combination with cyclic immunofluorescence for ETGs and other proteins regulated by ERK, including the canonical ETGs Egr-1, Fra-1, c-Jun, c-Myc, c-Fos, and phosphorylated proteins such as pERK, pc-Fos, and pRb (a downstream marker of ERK-dependent cell cycle entry; for convenience we collectively refer to all of these markers as ETGs). Because of the well-established interdependence of protein stability and ERK-mediated phosphorylation found in these ETGs, we hypothesized that this panel would provide the best assessment of ERK activity available through immunofluorescence.”

2. Related, if the model should be applied to complex tissue samples, the question of which

ETG to use and which ETGs would be expressed and with which dynamics has not been addressed, even speculatively.

We have added a paragraph to the discussion to speculate on which combinations of ETG stains could provide useful. It reads:

“Nonetheless, our data support the utility of multiplexed immunofluorescence datasets for capturing cell signaling states via histological methods. For example, Fra-1 is consistently correlated with long-term ERK activity, while Egr-1 and pERK correlate on the short term, across all models tested. Combining these stains could distinguish cells with sustained ERK activity (which is found in RAS mutants⁴⁰) or cells with highly stochastic ERK activity (found in receptor-driven cancer models^{11,29}) from cells in which ERK is activated in a regular pattern (which has been found for non-tumor tissue⁴²). It is also notable that while all three cancer cell lines have lost the correlation between pRb and Fra-1 that is present in non-tumor cells, they have retained the correlation between pRb and c-Myc (Fig. S8). Such changes have implications as indications of targetable tumor-specific pathway dependencies, and as potential signatures of signaling rewiring that could be readily detected in histological samples. The disruption of c-Jun, Egr-1, and Cyclin-D1 regulation in cells with B-Raf mutations suggests additional such signatures that could be detected^{27,43}. Furthermore, the coherence between ETGs within the same cell, a factor not explicitly explored here but available within our datasets, can likely provide indications of transformed signaling activity. Thus, our study supports the feasibility of inferring activity dynamics from fixed cell immunofluorescence.”

3. Fig 5 A and B- cluster 4 and 5 appear mislabeled.

We thank the reviewer for this observation. This error has now been corrected.

Reviewer #3 (Remarks to the Author):

In this study, Ram et al. investigated whether the history of ERK activity dynamics can be predicted from the expression levels of ERK target genes by analyzing ERK activity dynamics with FRET imaging and cyclic immunofluorescence. First, the authors added different concentrations of EGF and/or MEKi to EKAR3.5-expressing MCF-10A cells, fixed the cells at the endpoint, and obtained expression levels of pERK, Fra-1, c-Myc, and others using 8 different antibodies (Fig. 1). Using these large amounts of data, they investigated the relationship between ERK dynamics and ETGs by correlation (Fig. 2) and regression analysis (Fig. 3). Furthermore, using CNNs, they predicted the expression of ETGs based on the dynamics of ERK activity (Fig. 4), predicted the dynamics of ERK activity based on the levels of ETGs using a decision tree model (Fig. 5), and analyzed the potential dynamics of ETGs using an ERK-dependent gene expression model (Fig. 6). These analyses show that Fra-1 and pRB levels reflect the duration of ERK activity, while Egr-1 and c-Myc indicate the recent level of ERK activity. The authors also suggest the possibility of predicting ERK activity dynamics from multidimensional immunostaining in the future.

Overall, the work is, for the most part, sound, and the experiments are well-designed. I really enjoyed reading this excellent piece. I provide some comments below that would be worth addressing in a revised version.

1. The data presented here are mainly analyzed for the dynamics of ERK activity and correlation analysis with ETGs. However, the data also include spatial information, but with the exception of Figs. 2d, 2f, and 5e, the spatial information is mostly ignored. In fact, several groups have reported that ERK activity propagates to neighboring cells. Can the authors show whether they can predict ETGs more accurately by using the ERK activity dynamics of neighboring cells, and whether they can predict ERK activity dynamics more accurately by using the information on the ETGs of neighboring cells? The spatial transcriptome is currently available, and I thought that it would make the significance of this paper stand out even more.

We thank the reviewer for their helpful comments and critiques.

We now include another analysis in Figure 6 where we account for spatial ETG information from neighboring cells. In this analysis, we used ETG information from hexagonal regions of the monolayer, to predict ERK activity. We found that this method greatly improved the prediction models, increasing from ~60% prediction accuracy to ~80% accuracy for MCF10A cells (Fig. 6e). Furthermore, these improvements were consistent across the new cell lines added to the study.

2. Line 99, Figure S1a-d: The authors should clarify the mode of action and meaning of calibration of EKAR3.5. First, it should be stated in the text that EKAR3.5, like other ERK biosensors, measures the balance between phosphorylation by ERK and dephosphorylation by phosphatase. Second, there is no rationale for the FRET measurement to be linear with respect to the phosphorylation level of EKAR3.5 in Figure S1d. Perhaps it should be fitted to the Hill function, in which case more data points would be needed, but I am not going to ask the authors to do this. Further, if it was a linear fit, then the calibration essentially has no effect on the later analysis at all. Essentially, FRET measurements should be correlated with pERK levels, but this is also outside the scope of this paper. In any case, please add an explanation of the mode of action of EKAR3.5 and the meanings of calibrations so that readers do not misunderstand.

We have updated the introduction of the sensor for clarity. It now reads, “In MCF10A mammary epithelial cells, we used EKAR 3.1, a calibrated FRET-based biosensor that measures the balance between phosphorylation by ERK activity and dephosphorylation by competing phosphatases”.

The analysis of the EKAR FRET sensor and the development of its calibration were performed previously, via mathematical models of the reporter and of epifluorescence imaging systems. To make this clearer for the readers, we have updated the text in our Methods that discusses the EKAR 3.1 calibration. We now make it clear that the “EKAR 3.1 signal” is preprocessed, and provide a quick view of the processing function, while the complete derivation is available via reference.

Additionally, please note that we have decided to refer to the version of the EKAR reporter we used as “EKAR3.1” rather than “EKAR3.5”, which we think better reflects its position within the lineage of FRET-based ERK reporters (as the first update of EKAR3). This specific version has not been published elsewhere, and we have updated all instances of its name in the paper.

3. I presume that the response function of ERK activity dynamics for each ETGs by linear regression model would further explain clearly the authors' claim that Fra-1, pRB are induced by sustained ERK activity and EGR-1, c-Myc by transient ERK activity.

We agree that dynamic response functions, such as a time course of ETG expression after a step or impulse of ERK activity, would provide a complementary view of the differences we observe. However, directly measuring such responses is outside the scope of this study, as it would require live-cell methods of tracking ETGs. We have explored this connection to some extent in previous studies (Gillies et al., 2017, ref. 19 and Davies et al., ref. 29), which do bear out the reviewer's suggestion that ETG response functions are consistent with their timescale of correlation.

4. Line 280: I did not understand what the authors wanted to show in Figure 5e. Did they classify ERK activity dynamics from immunostaining data and map it to cells? Or did they acquire new data and validate it? Could you please add some more explanation?

In this figure (now Fig. 6f), we indeed classified the ERK dynamics based on immunostaining and mapped the classifications to the corresponding cell image (from the final time point). Our intent was to demonstrate an application of how these statistical models can be applied to create useful visualizations that can be analyzed for further insight on the spatiotemporal histories of a monolayer. We have rewritten this section and updated the figure to make the purpose and methodology clearer.

POLICIES AND FORMS REQUIRED FOR RESUBMISSION

* Please complete or update the following checklist(s) to verify compliance with our research ethics and data reporting standards. Address all points on the checklist, revising your manuscript in response to the points if needed.

The form(s) must be downloaded and completed in Adobe Reader rather than opened in a web browser. Each form must be uploaded as a Related Manuscript file at the time of resubmission.

* Editorial policy checklist:

<https://www.nature.com/documents/nr-editorial-policy-checklist.pdf>

>We have included the checklist and verified that the manuscript has been updated accordingly.

* Reporting summary:

>The reporting summary has been updated for the revised manuscript.

* As Nature Portfolio policies strongly encourage you to share your research data in a public repository (e.g. spreadsheets, text, images), we are partnering with the figshare repository so that you can use the figshare integration via the 'Research Data Deposition' tab when submitting your revised manuscript.

Data are stored privately until a manuscript decision is reached and you can edit/withdraw them up to this point: you retain rights and control over your data. The data will be published at the same time as your article; you will receive a data DOI, with guidance on linking the data and manuscript. In the event your manuscript is not accepted, you can keep or remove your data in figshare.

We recommend the use of discipline-specific repositories where available and for a number of data types this is mandatory. Ensure you do not submit these data types or any sensitive data to figshare.

>We have uploaded all of our datasets and all associated code needed to generate models and figures to figshare.

* Your work characterises chemical or biomolecular materials. Please see the link below for reporting requirements. There is no form to upload but you may need to revise your manuscript to comply with this policy.

<https://www.nature.com/ncomms/submit/chemical-characterisation>

>Data validating the biological construct EKAR3.1 (a genetically encoded reporter for ERK activity), and cell lines expressing the reporter, are provided within the manuscript.

>* Your paper uses custom code/software. Please complete the following code and software submission checklist and make your code available for reviewer assessment, if you have not already done so. The code/software can be provided in a zip file with a readme.txt file or other instructions for installing and running the software. If appropriate, also provide example data and expected output. If you have any issues with the file upload, please let me know.

<https://www.nature.com/documents/nr-software-policy.pdf>

>All custom code is included in our figshare repository along with instructions.

DATA AND CODE AVAILABILITY

* All Nature Communications manuscripts must include a "Data Availability" section after the Methods section but before the References. If any of the data can only be shared on request or are subject to restrictions, please specify the reasons and explain how, when, and by whom the data can be accessed. For more information on this policy and a list of examples, see:

<https://www.nature.com/documents/nr-data-availability-statements-data-citations.pdf>

* Please also include a “Code Availability” section after the “Data Availability” section. If the code can only be shared on request, please specify the reasons. For more information on our code sharing policy and requirements, please see:

<https://www.nature.com/nature-portfolio/editorial-policies/reporting-standards#availability-of-computer-code>

>We have included this section in the manuscript.

* We strongly encourage you to deposit all new data associated with the paper in a persistent repository where they can be freely and enduringly accessed. We recommend submitting the data to discipline-specific and community-recognised repositories; a list of repositories is provided here: <http://www.nature.com/sdata/policies/repositories>

Refer to our data policies here: <https://www.nature.com/nature-portfolio/editorial-policies/reporting-standards#availability-of-data>

>All data have all been included in the figshare repository.

* To maximise the reproducibility of research data, we strongly encourage you to provide a file containing the raw data underlying the following types of display items:

- Any reported means/averages in box plots, bar charts, and tables
- Dot plots/scatter plots, especially when there are overlapping points
- Line graphs

The data should be provided in a single Excel file with data for each figure/table in a separate sheet, or in multiple labelled files within a zipped folder. Name this file or folder ‘Source Data’, and include a brief description in your cover letter. The “Data Availability” section should also include the statement “Source data are provided with this paper.”

To learn more about our motivation behind this policy, please see:

<https://www.nature.com/articles/s41467-018-06012-8>

>We have provided source data files through figshare that contain all data depicted in the manuscript, along with code to regenerate the figures. We have noted this in the data availability statement.

*We also mandate the presentation of uncropped versions of any gels or blots, labelled with the relevant panel and identifying information such as the antibody used.

>Uncropped and labelled versions of western blots are included in the Supplementary Information file as Figure S11.

* Please replace your bar graphs with plots that feature information about the distribution of the underlying data. All data points should be shown for plots with a sample size less than 10. For larger sample sizes, please consider box-and-whisker or violin plots as alternatives. Measures of centrality, dispersion and/or error bars should be plotted and described in the figure legend.

>All bar graphs have been updated to include distribution data or individual data points, as appropriate.

NCOMMS-23-05536B Response to Reviewer Comments

Reviewer #1 (Remarks to the Author):

The authors responded well to my questions in the previous submission, alleviating my concerns about the R2 for prediction accuracy.

I have not found the source codes related to the manuscript for further verification. Specifically, multiple programming languages and algorithms have been adopted. Sharing instructions on data processing with source codes will help readers better understand the results.

Reviewer #1 (Remarks on code availability):

No code is uploaded.

We thank the reviewer again for their helpful critiques. We apologize that the code was not easily accessible. While our code was included in the figshare repository created for this paper, we realized only recently that because the code was bundled with the raw data, the download time from figshare was several hours. To alleviate this problem, we have now split the figshare repository into separate code and data files. The code file should now be readily accessible from this link:

<https://figshare.com/s/f3f994d32dbda8f48101>

Reviewer #3 (Remarks to the Author):

The authors have done a mostly adequate job of responding to the reviewers' comments.

We thank the reviewer again for their helpful comments and suggestions.

Reviewer #4 (Remarks to the Author):

This timely study characterizes the ERK signaling dynamics associated with specific and distinct effector outputs. It is a resource for the signaling community and timely in terms of shedding light on which effectors might best be used clinically to test for residual ERK activity. The work is quite comprehensive, but could still be improved by correlating the effector output with specific cell phenotypes. Specific comments are below.

We thank the reviewer for their evaluation of our manuscript and the helpful suggestions.

1. The interpretation of the EGF treatments (different concentrations and durations of treatment) in Figure S3 is limited by the focus on only EGF. In vivo, cells are exposed to many different growth factors at the same time and rarely experience a complete starve followed by a single growth factor stimulation. The authors try to address this more physiological approach by including stimulations in the presence of serum in figures S5-7. While this does not completely address the question of how translatable the current data and model are to cells in vivo, it is a good first step towards what will likely be a future long-term effort to develop this approach.

We agree that including additional stimuli, which are known to create substantially different ERK kinetics, would be a very valuable addition. We do hope to explore this in future studies. To highlight important gaps yet to be addressed, we have commented on the need to examine additional stimuli and pathways in the Discussion section (lines 508-515).

2. The study in 3 different cancer cell lines is very interesting and key to the generalizability and utility of the conclusions. The discussion and overall conclusions from this expanded study should be improved. The genetic drivers that affect ERK and other key cell proliferation, survival, and migration pathways should be more clearly called out (HCC827 – EGFR, A549 – KRAS, MCF7 – PI3KCA) and put in context with how ERK is known to signal in the presence of these mutations, work by this lab and others. The lack of ETG-ERK activity correlations in the MCF7 line may be due to its dependence on PI3K/AKT signaling, rather than ERK.

We have now emphasized the genetic differences between the cell lines in both the Results and Discussion sections. We have also provided additional discussion of the effects of these mutations on the signaling behavior in the Discussion section (lines 475-491). We have also included a note on the suggested hypothesis that MCF7 have a greater loss of ETG-ERK activity due to their PI3K dependence. Thank you for this suggestion.

3. The conclusions from this extensive 4 cell line, multiple agonist study should be better summarized for interpretation of the biological significance of key findings through a few possible approaches:

a. The UMAP approach in Figure S5-7 could be applied so that different colors show the individual ETGs, rather than the replicates. If they current presentation, they provide

little new information. Do feature populations fall out that explain the different correlations with respect to the cells' driver mutations and dependencies.

We have now added UMAPs colored by seven of the ETGs for all cell lines; for MCF10A they are shown in Fig. S2f, and for the cancer cell lines in Figs. S5e, S6e, and S7e. These are useful visualizations, and in general, the subpopulations observed in these visualizations correspond to the correlations shown in Fig. S8, including the association of ETGs with pRb found in MCF10A but not in the 3 cancer cell lines. For MCF10A, the pRb-positive cells overlap more closely with the ETG stains than they do in the cancer cell lines.

b. Since the relevant ERK signaling features appear to be different in the different cell lines, it is important to determine if the ERK signaling and ETGs are also different for the cell phenotype.

Does inhibiting ERK have a bigger effect on the proliferation of HCC827 and A549 cells and a relatively small effect on MCF7 cells? Is this dependent on Fra-1 and/or myc?

We have addressed this question by calculating the number of pRb-positive (i.e., actively cycling) cells following MEK inhibition for each of the cell lines. We find that all three cancer cell lines are insensitive to MEK inhibition, while MCF10A are sensitive. While MCF7 show more loss of correlation than HCC827 or A549, all three show evidence of reduced correlation. In particular, we note that all three cancer lines have a low Fra-1 correlation and high c-Myc correlation. Thus, the loss of dependence of proliferation on ERK signaling is not proportional to the degree of overall ERK-ETG correlation but does match the lack of Fra-1-pRb correlation. Our interpretation is that driver mutations such as Ras or EGFR are initially responsible for driving proliferation early in tumorigenesis but become uncoupled from downstream targets later in the evolution of the tumor, which is consistent with previous studies (PMID: 17450133). We include a discussion of these features on lines 508-524 of the Discussion.

Does c-Jun correlate better with ERK signaling in A549 cells because the other pathways controlling c-Jun are not active in these cells?

This is an interesting hypothesis. Examination of the c-Jun responses indicates that all of the cell lines show increases in c-Jun after EGF stimulation, but this happens to a higher degree in A549 cells. This result could occur as a result of different patterns of MAP3K expression, as members of the fairly large MAP3K family display different biases toward ERK vs. JNK (Peterson et al. 2022, PMID 36356576). We don't have additional data to confirm this idea and have chosen not to speculate on it in the text.

What is the meaning of Egr-1 correlation with ERK pulse frequency?

Because Egr-1 acts as a negative regulator of its own transcription, it is expressed most highly when ERK activity is present as intermittent pulses (Davies et al. 2020). Our frequency metric is defined as pulses per time, and thus the highest values of frequency correlate with the best conditions for Egr-1 expression.

Does this suggest an upcoming mitosis, while Fra-1 controls a baseline level of mitotic capability?

The timing of pulses that stimulate Egr-1 expression does not appear to be tightly coupled to upcoming mitotic events, as pulses can occur throughout the life of the cell independently of the cell cycle (for example, see Fig. 2a; low pRb cells toward the bottom of the heatmap still have ERK activity pulses).

Understanding what the key features are that suggest meaningful ERK activity will be important for using these readouts in assessing pathway targeting in cancer treatments.

We completely agree, and we have done our best to highlight this perspective in the substantially rewritten discussion. We hope that by providing this unique dataset in its entirety, we will enable further research into the new signaling features that are detectable at this scale.